# Direct observation of a crescent-shape chromosome in expanded *Bacillus subtilis* cells

Miloš Tišma [1], Florian Patrick Bock[2], Jacob Kerssemakers[1], Hammam Antar[2], Aleksandre Japaridze[1], Stephan Gruber[2] & Cees Dekker[1] ✉

Bacterial chromosomes are folded into tightly regulated three-dimensional structures to ensure proper transcription, replication, and segregation of the genetic information. Direct visualization of chromosomal shape within bacterial cells is hampered by cell-wall confinement and the optical diffraction limit. Here, we combine cell-shape manipulation strategies, high-resolution fluorescence microscopy techniques, and genetic engineering to visualize the shape of unconfined bacterial chromosome in real-time in live *Bacillus subtilis* cells that are expanded in volume. We show that the chromosomes predominantly exhibit crescent shapes with a non-uniform DNA density that is increased near the origin of replication (*oriC*). Additionally, we localized ParB and BsSMC proteins – the key drivers of chromosomal organization – along the contour of the crescent chromosome, showing the highest density near *oriC*. Opening of the BsSMC ring complex disrupted the crescent chromosome shape and instead yielded a torus shape. These findings help to understand the threedimensional organization of the chromosome and the main protein complexes that underlie its structure.

Over the past decade, it has become evident that bacterial chromosomes are folded into a compact 3D architecture that regulates transcription and is necessary for cell survival[1–3]. As nuclear compartmentalization is absent in bacteria, a high abundance of DNA-binding proteins can locally bind and change the chromosomal structure[4]. Furthermore, the chromosome is subject to the crowding effects of cytosolic components and the genome has to fit within the confines of the cell boundary which squeezes the millimeter-long bacterial genome within the micron-size cell[5]. Although bacterial chromosomes have been imaged by microscopy in many earlier works[6–11], the tight confinement of the genome combined with the finite optical diffraction limit has been a hindrance toward resolving its unconfined shape and structure. Attempts to significantly increase the spatial resolution via super-resolution microscopy techniques often require synthetic dyes that include lengthy washing procedures or

invasive crosslinking steps which may affect the DNA structure and dynamics in live cells[6,7,12,13]. Direct high-resolution imaging of the chromosome would shed light onto the chromosomal dynamics, macrodomain organization[14,15], and the roles of various proteins in its organization.

An indirect but powerful method of studying the chromosomal structure is provided by chromosome capture techniques (3C[16], 4C[17], 5C[18]), most notably 'Hi-C'[7,19–24]. This technique uses proximity-based ligation in combination with next-generation sequencing to uncover the average spatial organization of the DNA within a population of cells[21]. Over the past decade, Hi-C has been established as a major tool for studying the 3D chromosome organization and it has contributed key insights about the chromosome organization in many bacterial species[7,19,20,23–25]. Interestingly, the most well-studied bacterial model organisms *Escherichia coli* and *Bacillus subtilis* show an entirely

[1]Department of Bionanoscience, Kavli Institute of Nanoscience Delft, Delft University of Technology, Delft, Netherlands. [2]Department of Fundamental Microbiology (DMF), Faculty of Biology and Medicine (FBM), University of Lausanne (UNIL), Lausanne, Switzerland. ✉e-mail: C.Dekker@tudelft.nl

different chromosomal structure in Hi-C maps. *E. coli* was deduced to have a textbook circular chromosomal shape with separately condensed chromosomal arms that are twisted within the cylindrical cell[19]. *Bacillus subtilis* however, shows a distinct "second diagonal" feature in the Hi-C maps which indicates that two chromosomal arms are spatially aligned, which presumably is facilitated by the action of bacterial structural maintenance of chromosome (SMC) proteins[7,23,24]. Aligned chromosome arms were also observed in other microorganisms[20,26–28], albeit for some (e.g., *Caulobacter crescentus*), the second diagonal did not span the full length of the chromosome, suggesting a segregation of the nascent origins but not a full zipping of the entire chromosome[20]. Many questions remain, however, as Hi-C methods involve extensive cell fixation which can alter the DNA organization[21], and conclusions are drawn from population averages where the single-cell structure and cell-to-cell variabilities are lost.

A promising way to spatially resolve the bacterial chromosome is to use cell-shape manipulation techniques[15,29–32] or expansion microscopy (ExM)[33–35]. While expansion microscopy, where bacteria are encapsulated in a hydrogel that is stretched, requires cell fixation that precludes live-cell imaging[35], cell-shape manipulation techniques allow for imaging of live cells, where a single chromosome can be maintained via DNA replication halting. Using cells-shape manipulation approach, we previously visualized the *E. coli* chromosome at the single-cell level, showing a toroidal shape with fast local dynamics and interesting substructures within the genome[15,29].

Here, we use cell-shape manipulation with replication halting to capture the structure of the unconfined *Bacillus subtilis* chromosome in single live cells. Cell-shape manipulation relaxed the cell-boundary confinement of the chromosome in live bacteria, thus allowing standard super-resolution microscopy techniques to capture the shape of the chromosome without confinement by the cell-wall boundary. We observed that the unconfined *B. subtilis* chromosome most frequently adopts a crescent shape, with an origin region at one tip of the crescent that is highly condensed. Furthermore, we measured the positioning of the key chromosome-organizing proteins ParB and BsSMC along the contour of the chromosome. Upon BsSMC disruption, the crescent chromosome shape changed into a torus shape, indicating that BsSMC is required for maintaining the crescent shape. The data provide insight into the chromosomal organization of the *B. subtilis* chromosome and its organizing proteins.

## Results

### Cell-shape manipulation of *Bacillus subtilis* bacteria

In standard growth conditions, both for nutrient-rich or in a minimal medium, *B. subtilis* cells grow into rod shapes with its chromosome residing close to the center of the cell as a confined object of ~1.3 μm × 0.7 μm (Fig. S1A, B)[5]. Due to the diffraction limits of microscopy, most of the chromosome's inner structure cannot be resolved. To eliminate the confining effects set by the cell wall, we converted the rod-shaped *B. subtilis* cells into spheroidal cells (also known as L-form cells[36,37]) similarly to the protocol described in Kawai et al.[38]. (Fig. 1A, B, see "*Methods*" section). The physical conversion from a capped cylinder to a spheroid implies a change in surface-to-volume ratio. In our case, the total surface remains constant as it is set by the cell membrane that cannot substantially grow over the course of the fast conversion to spheroidal cells. To accommodate the change and increase the cell volume, we used different osmotic media to promote water uptake into the spheroidal cells, and accordingly a volume increase. We tested a variety of media ranging from highly hypoosmotic (100 mM osmolyte) to isoosmotic concentrations (500 mM osmolyte, see "*Methods*" section), which did not affect the phenotype in cylindrical cells (Fig. S1C). Under wall-less conditions, we observed the highest increase in volume under slightly hypoosmotic conditions, viz., 300 mM osmolyte (Figs. 1C, S1D). Cylindrical cells of, on average, $3.3 \times 0.8 \times 0.8$ μm³ size then adopted an approximately spheroidal shape with an average size

of $2.3 \times 2.3 \times 1.7$ μm³. This resulted in the average total volume change of up to about a factor of 3 of the original volume. This volume expansion increased the physical space where the chromosome can freely reside within a few minutes, allowing for observation of its intrinsic shape without confinement while avoiding long cell-reshaping treatments which could induce chromosome perturbations.

### Observation of a crescent chromosome *in Bacillus subtilis* in single cells

To obtain the image of a single chromosome in individual cells, we used two replication-halting strategies in separate strains. First, we constructed a strain containing temperature-sensitive DnaB protein (*dnaB134ts*, Fig. S2, Table 1)[9,39,40]. This protein is an essential component during the DNA replication process in *B. subtilis* as it facilitates new rounds of initiation by facilitating helicase loading[41,42]. Upon a temperature increase, the strain carrying DnaB[K85E] (*dnaB134ts* locus) experiences a strong inhibition of initiation of new DNA replication rounds[40]. To control for the possible denaturation of HbsU protein at high temperatures[43] (which would alter chromosomal structure), we grew the strain at a maximum of 39 °C. Second, we constructed a separate strain carrying $P_{lac}$-*sirA* whose protein product can also efficiently halt the initiation of DNA replication by directly inhibiting the action of DnaA[44,45] (Table 1), whilst preserving HbsU stability at 30 °C[43]. For DNA imaging, we used (i) a fluorescently labeled version of the HbsU protein which uniformly binds across the DNA[46,47], or (ii) synthetic DNA intercalating dyes (Fig. S2) that do not affect cell viability[48], or (iii) standardized DNA dyes (DAPI, SYTOXGreen, SYTOXOrange). All different fluorophores and replication-halting variants yielded similar results (Fig. S2).

In a large fraction (33–43%) of the cells, we could resolve the unconfined chromosome shape even under wide field microscopy and observed that the *B. subtilis* chromosome exhibited a crescent shape (Figs. 1D, S3A–D). This was observed irrespective of the chromosome labeling or replication-halting strategy (Fig. S2).

Interestingly, the size of the expanded chromosome in *B. subtilis* was smaller than the cell diameter (Fig. S4), which indicates that the chromosomes did largely expand to their intrinsic conformation. This contrasts our previous observations in *E. coli*, where chromosomes that assume a toroidal shape upon cell expansion, fully expanded to the cell size[15]. A considerable cell-to-cell variability was observed where some chromosomes appeared as a more condensed object that we were unable to resolve (Fig. S3C, D).

Next, we used super-resolution microscopy to obtain high-resolution images of single *B. subtilis* chromosomes. Deconvolution of wide-field microscopy and structured illumination microscopy (SIM) allowed for the imaging of live cells with spatial resolution down to 150 nm and 120 nm, respectively. Images taken with both techniques consistently showed that chromosomes adopted a crescent shape (Figs. 1E, S5, Movies S1, S2). This is consistent with previous Hi-C data that suggested a close spatial alignment of left- and right-chromosomal arms[24]. Interestingly, the fluorescence intensity along the contour of the chromosome appeared to be quite variable between two ends as well as from cell to cell. This suggests a dynamic compaction level of the DNA along the chromosome where DNA is distributed non-uniformly (Fig. 1D, E). To quantitatively measure the physical characteristics of the crescent-shaped chromosomes, we ensured the presence of a single chromosome within each cell, as any unfinished replication round could result in large changes in the DNA amount and chromosome shape, thus obstructing quantitative assessments of the chromosome shape, size, and dynamics.

### DNA is highly compacted in the origin region

First, we located the origin of replication (*oriC*) as the reference point. Conveniently, *B. subtilis* has multiple *parS* sites around the *oriC*, which bind partitioning protein B (ParB) in high numbers[49] (Fig. 2A). ParB proteins bridge multiple *parS* sites and form a single focus per

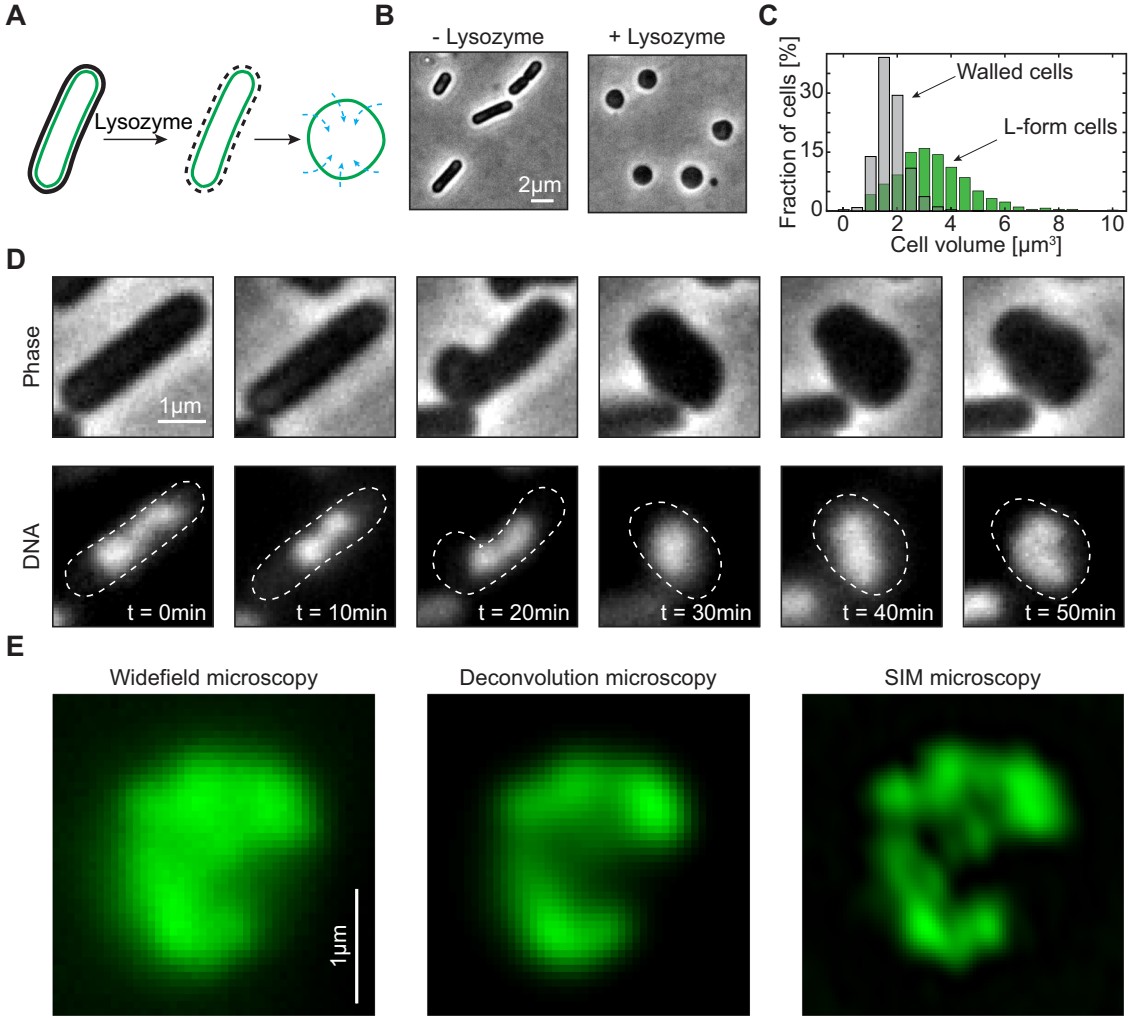

**Fig. 1 | *Bacillus subtilis* chromosome adopts a crescent shape upon cell widening. A** Graphical representation of the cell-shape conversion from rod-shaped to spheroidal shape. **B** Conversion of rod-shaped cells to spheroidal cells under hypoosmotic conditions in SMM + MSM medium (300 mM succinate) upon addition of lysozyme. **C** Total cell volume before (gray, $V_{avg} = 1.8 \pm 0.6\,\mu m^3$, $N = 550$) and after the lysozyme treatment (green, $V_{avg} = 3.4 \pm 1.5\,\mu m^3$ $N = 1056$). The results are pooled from three biological replicates. **D** Example timelapse imaging of the conversion of a single cell from rod shape to spherical shape under an agar pad for the BSG217 strain at 39 °C. **E** Comparison of wide field (left), deconvolved (middle), and Structured Illumination Microscopy (SIM) (right) image of a crescent chromosome in the strain BSG217. The experiments were performed in duplicates with similar results.

chromosome[50,51], and they are commonly used as a proxy for the number of chromosomes per cell[24,52–55], given that newly replicated origins are quickly separated. We used a fluorescent fusion of ParB-mScarlet in addition to $P_{lac}$-*sirA* in order to select for cells with a single fluorescently labeled origin-proximal region[56] (Fig. 2B–E, Table 1). In growth assays, this strain behaved equivalent to wild-type *B. subtilis* (Fig. S6A) and it showed the same phenotype under the microscope (Fig. S6B), while experiencing severely halted growth during the SirA induction (Fig. S6C). In the absence of SirA induction, this strain showed multiple origins per single cell (Fig. 2B, C). Upon induction of SirA protein, up to 78% of the cells showed a single focus per cell, indicating successful replication halting (Fig. 2D, E). To ensure that this was not the effect of unresolved origins of multiple chromosomes, we performed a qPCR to quantify the *ori:ter* ratio and further constructed a strain that had fluorescently labeled replication termination protein (RTP-GFP) in addition to the origin label. By quantifying the *ori:ter* ratio in the cells upon replication halt using both qPCR and fluorescent readout, we confirmed that SirA expression indeed strongly suppressed replication due to the replication halt treatment (Fig. S7).

In origin-labeled cells, we observed *ori* to localize at the tip of the crescent chromosome (Figs. 2F, S8). The origin of replication was not

notably mobile, residing within 86 nm (Fig. S9A–C) on the time scales of 10 s, which is comparable to the minor background movements of our cells under the soft agar (FWHM = 59 nm, Fig. S9D). Cells that had both *ori* and *ter* labeled, showed that these two important loci tend to localize at the opposite tips of the crescent-shaped chromosome (Fig. S10), albeit with a high flexibility in the *ter* region that induced significant fluctuations of the ter position away from the tip (Fig. S1E).

After thus successfully localizing the *oriC* and selecting for the presence of a single fluorescent focus (i.e. a single chromosome), we observed the fraction of crescent chromosomes to rise to 58% indicating that this is the most frequently observed shape in cells with a single chromosome (Fig. S11). Thus having mostly single crescent-shaped chromosomes in our expanded cells, we proceeded with the analysis of DNA compaction along the contour of the crescent chromosome (Fig. 2G, see "Methods" section and Wu et al.[15]). To stain the DNA we used either fluorescent intercalator dye – SyG (Fig. 2F–J)[57] or DNA-binding protein fusing HbsU-mTurquoise2 (Fig. S12A–C), allowing us to correlate the fluorescence intensity with the underlying amount of DNA. In all cells, we observed an increase in the DNA fluorescence intensity proximal to the ParB focus, i.e., at the origin of replication (Figs. 2H, I, S12D, E). We observed an uneven distribution of

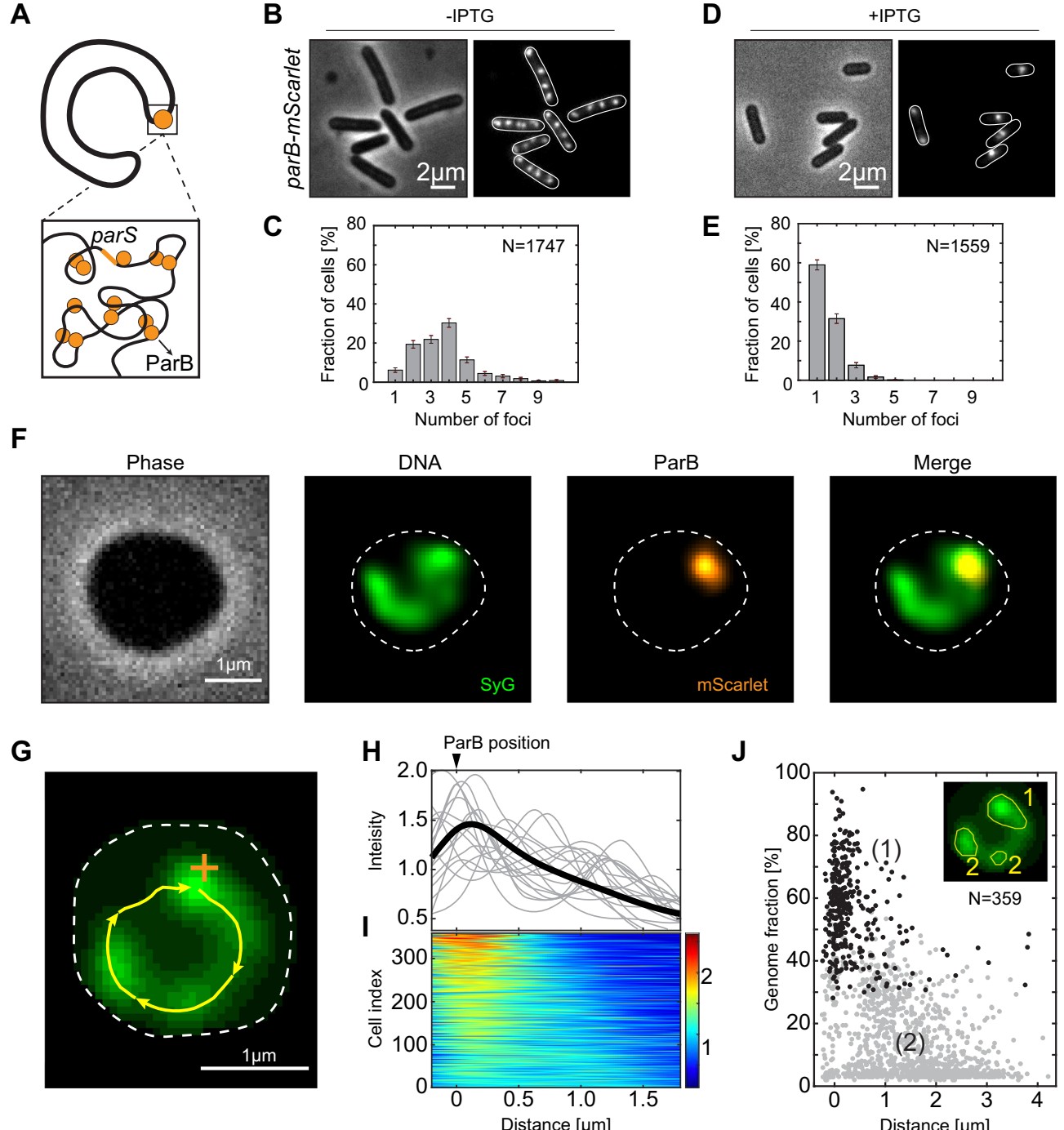

**Fig. 2 | ParB complex co-localizes with a region of high DNA density. A** Graphical representation of the crescent-shape chromosome. The zoom sketches the origin/*parS* region that is condensed by ParB proteins (orange). **B** Phase (left) and fluorescence (right) images of the BSG4595 cells (*parB-mScarlet*; $P_{hyperspank}$-*sirA*) in the absence of ITPG. **C** Quantification of the number of fluorescent ParB-mScarlet foci in single cells imaged as in (**B**) (*N = 1747*, avg$_{spots}$ = 3.56 ± 1.8). The results are pooled from three biological replicates. Error bars represent 95% confidence intervals around the obtained value. **D** Same as (**B**) but for cells in the presence of 2 mM ITPG for 150 min. **E** Quantification of the number of fluorescent ParB-mScarlet foci in single cells imaged as in (**D**) (*N = 1559*, avg$_{spots}$ = 1.42 ± 0.86). The results are pooled from four biological replicates. Error bars represent 95% confidence intervals around the obtained value. **F** Representative example of a cell after treatment with lysozyme (400 μg/ml) for 30 min. The experiments were performed at least three

times (triplicates) with similar results. **G** Crescent-shaped chromosome. Cell outline is denoted by the white line; DNA contour track by yellow line; location of the ParB focus by the orange cross. **H** DNA intensity along the contour of the chromosome (cf. yellow line in **G**). Black line shows the average normalized intensity obtained from all cells (*N* = 359). Gray lines display arbitrarily chosen individual examples. The position of the ParB focus is indicated on top (defining the 0 μm position). **I** DNA density along the chromosome in all individual cells. Colorbar represents the fold-increase. Cells are ordered from top to bottom in terms of contrast. **J** Cluster analysis of the chromosome. Inset shows an example output of the analysis with 1 primary and 2 secondary-size clusters. The fraction of the genome that is contained within the primary focus (1) (black dots) is plotted versus distance from the origin (ParB focus; *n* = 359). Secondary condensation (2) foci are represented as gray dots.

DNA along the contour of the chromosome, with a higher abundance of DNA present near the origin of replication and less DNA present toward the tail of the crescent chromosome. Using previously established analyses[15], we were able to detect DNA clusters along the chromosome based on the local fluorescence and intensity changes, where clusters were detected as a sub-group of point spread function-limited spots (see "Methods" section for more detail). In our *B. subtilis* chromosomes, we observed clusters of DNA, representing more condensed regions along the contour of the chromosome (Fig. 2J). We termed the cluster with the highest amount of DNA as the 'primary cluster', and all other clusters found as 'secondary clusters' (Fig. 2J). We found that the primary clusters typically localized near the *ori* region (Fig. 2J – black data), i.e., coinciding with the ParB focus. Most of cells possessed a secondary condensed region, albeit variable in size and position along the chromosome (Fig. 2I – gray data). We did not observe a particularly preferred chromosome position for these secondary clusters, and such domains instead appeared to position at arbitrary locations along the chromosome contour shape as well as contained a variable amount of DNA.

As the primary clusters tended to be positioned near *ori*, we analyzed the total DNA percentage that was localized near *ori* (Fig. S8B, C). We observed that the majority of cells (67%) exhibited a very large fraction (>40%) of their entire genome within only 500 nm from the origin of replication. Our data thus show a high DNA condensation proximal to the origin region, and concomitantly lower DNA condensation along the rest of the chromosome.

## BsSMC proteins spread along the entire contour of the crescent chromosome

SMC proteins constitute key players in the large-scale chromosome organization in all domains of life[58]. For *B. subtilis*, this concerns the BsSMC complex[23,52,54]. Previous studies concluded that BsSMC is recruited to the origin of replication through interactions with ParB protein[52,54,59], whereupon it progressively "zips" the left and right chromosome arms together along the entire length of the chromosome[7,23,24] (Fig. 3A). We constructed a strain containing an origin label (ParB-mScarlet), DNA label (HbsU-mTurqoise2), and BsSMC label (BsSMC-mGFP) in order to co-visualize the chromosome along with ParB and BsSMC (Fig. 3B, Table 1).

We observed that SMC proteins positioned as a combination of typically 1–2 fluorescent foci with an additional clear signal that colocalized with the DNA signal over the entire chromosome (Fig. 3B)[52,54]. This starkly contrasts data for the ParB protein that showed that virtually all of the intracellular ParBs was captured into one focus, with very little background in the rest of the cell (Figs. 2B, D, S8)[60]. We quantified the distance of BsSMC-mGFP and ParB-mScarlet foci in rod-shaped cells, and observed a close proximity between the foci, i.e. an average mutual distance of only $126 \pm 28$ nm (mean ± std, Fig. 3C). Upon performing the same procedure in widened cells, we again observed the HbsU-labeled chromosome to adopt a crescent shape, with a ParB-mScarlet focus at the tip of the crescent-shaped chromosome (Fig. 3D). BsSMC-mGFP appeared to spread over the entire chromosome, with the highest intensity proximal to the ParB-mScarlet focus (Fig. 3D). In a control strain containing a tag only on BsSMC and not on ParB and HbsU, we observed the same distribution of BsSMC signal (Fig. S13).

We quantified the intensities of all signals along the contour of the chromosome (Fig. 3E) similar to those described in Fig. 2G–I. The average DNA signal, from HbsU-mTurquoise2, showed an increased condensation at the origin and a gradual decrease toward the terminus (Fig. 3F, G – left). The BsSMC-mGFP intensity, however, did not linearly scale with the DNA signal, but rather had a 1.6-fold increase at the origin compared to the average signal along the chromosome (Fig. 3F, G – middle), while the signal also gradually decreased toward the *ter*. A decreasing signal from origin to terminus can be expected if

BsSMC is loaded at *ori* by ParB[52,54,59] and removed at the *ter* by XerD[61], while BsSMC exhibits a finite rate to dissociate from the DNA. Such a distribution was captured previously by ChIP-seq[62], which was interpreted as stochastic unloading of BsSMC proteins along the chromosome arms[7,62]. In some cases, a secondary BsSMC focus was visible that did not correlate with the ParB location (Fig. 3G). ParB signal showed a pronounced peak near the tip of the chromosome and close to zero intensity signal along the rest of the contour length (Fig. 3F, G – right). Interestingly, while we did not observe that the intensity of ParB correlated with the amount of DNA within the main cluster (Fig. S14A), we did observe that higher SMC presence correlated with increased DNA content at the same site (Fig. S14B). This was true for both primary and secondary clusters, i.e., even away from the origin, an increase of SMC content correlated with an increase in the amount of local DNA (Fig. S14C).

## Disruption of the BsSMC complex opens the crescent-shaped chromosome into a toroidal shape

Since BsSMC has been postulated to serve as the main connection between the left and right chromosome arms[7,23,24], we next tested for a potential reshaping of the chromosome after disruption of the BsSMC complex. For this, we constructed a strain with HbsU-mGFP as the chromosome label but included previously described ScpA-TEV3[63] as well as $P_{xyl}$-*TEVp* (Table 1). ScpA is the kleisin subunit of the BsSMC complex that is a basic part of its ring-like structure, and it is essential in cells for fast growth[64]. The incorporated Tobacco Etch Virus (TEV) protease recognition site (TEV3) was within the ScpA protein (Fig. 4A), while the TEV protease itself was included at a different locus under a xylose-inducible promotor. This allowed us to controllably disrupt the BsSMC complex by opening the SMC-kleisin ring upon xylose addition, and subsequently observe the chromosome reshaping (Fig. 4A). We demonstrated high specificity of SMC disruption that only occurred in the presence of both the TEV3 cleavage site and xylose (Fig. 4B).

Expression of the protease resulted in dramatic macro-scale changes in the chromosome shape (Fig. 4C). Upon disruption of the BsSMC ring, the arms of crescent-shaped *B. subtilis* chromosomes opened. Sometimes, both arms separated entirely, resulting in a toroidal-shaped chromosome (Figs. 4C, S14). We observed that megabase-sized structural rearrangements of the chromosome occurred on the timescale of minutes, going from a well-defined crescent to a fully open state within ~30 min (Fig. S15A, B, Movie S3, S4). These changes were never observed in the absence of xylose, nor in a control strain lacking the ScpA-TEV3 recognition site (Fig. 4D, E, and Figs. S15, S16). We further questioned whether the crescent shape of the chromosomes is affected in the absence of ParB proteins where SMC complexes can possibly load along the chromosome unspecifically. To examine this, we constructed a *ΔparB* strain containing the inducible $P_{lac}$-*sirA* replication-halting system (see "Methods" section). This strain showed an entirely disturbed chromosome shape and not the standard crescent-shaped chromosomes. Most chromosomes adopted a toroidal shape, similar to SMC depletion, or a rather undefined shape where the DNA was spread throughout all of the cell (Fig. S17). The persistence of the toroidal shape over time indicated that there is a lateral compaction of the chromosomes even after the removal of SMC proteins. We hypothesized that lateral compaction of the chromosome might arise from a different source, namely transcription by RNA polymerase, which was recently proposed to underlie the bacterial chromosome structure in *E. coli* [65]. After treating the cells with rifampicin, a potent transcription-blocking agent, we observed that the chromosomes in the expanded cells lost their compact crescent-shaped structures and instead exhibited entirely dispersed or compacted structures (Fig. S18) – thus confirming this hypothesis.

The results show that BsSMC proteins are indeed the sole agents that link the two arms of the circular chromosome of *B. subtilis*, while transcription by RNAp is maintaining the lateral compaction of the

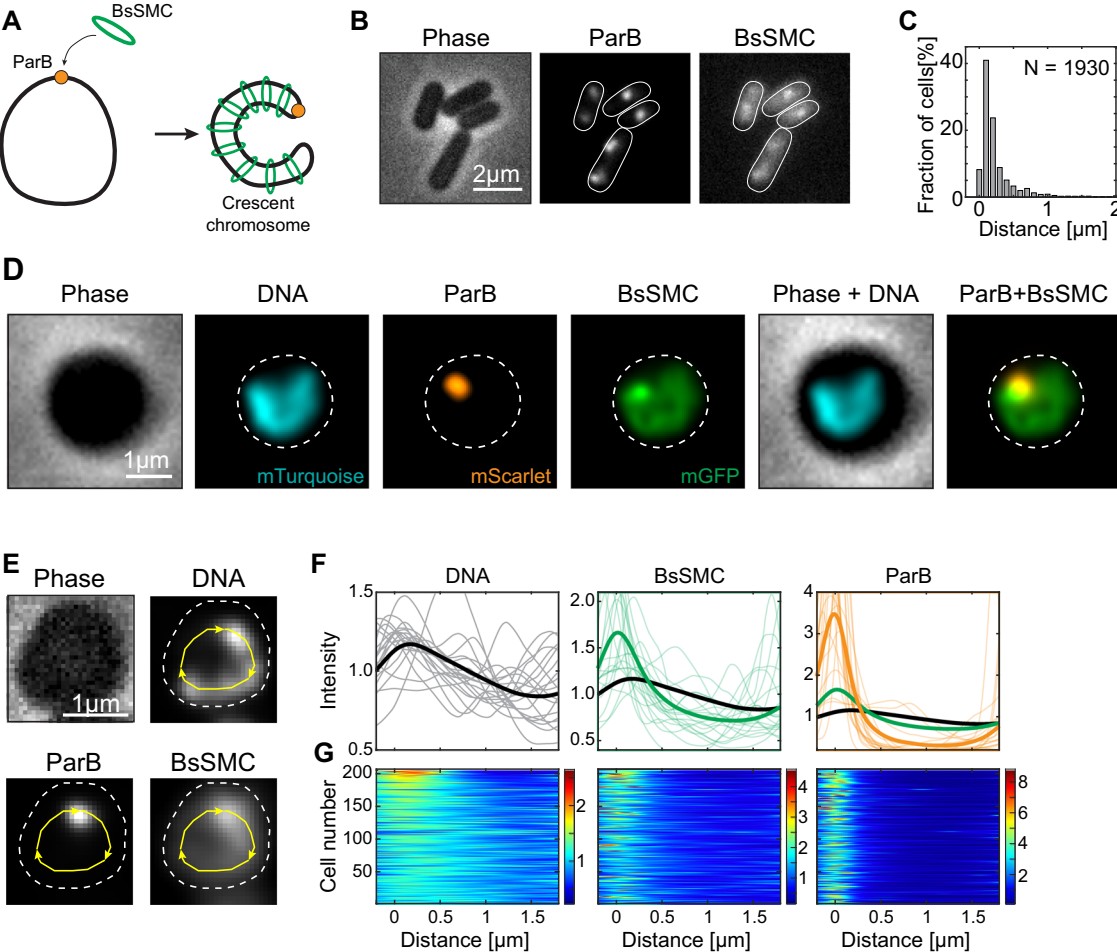

**Fig. 3 | BsSMC proteins spread along the entire chromosome. A** Graphical representation of the crescent chromosome and the BsSMC positioning (cf. Wang et al.[24]). **B** Phase (left) and fluorescent (right) images of BSG4623 strain (*hbsu-mTurqoise2; smc-egfp; parB-mScarlet; P_{hyperspank}-sirA*) after replication halt with 2 mM IPTG. **C** Histogram of the distance between the BsSMC and ParB foci. The average distance between spots was $d_{avg} = 126 \pm 28$ nm (mean ± std, $n = 1930$). The results for the histogram are pooled from three biological replicates. **D** Images of bacterial strain BSG4623 strain after lysozyme (400 µg/ml) treatment for 30 min.

Dashed white line represents the cell outline. The experiments were performed at least three times (triplicates) with similar results. **E** Contour analysis of DNA, BsSMC, and ParB signals along the crescent chromosomes (cf. Fig. 2G.) Dashed white line represents the cell outline. Experiments were performed in triplicates with similar results. **F** Fluorescence intensity along the contour line of the chromosome, starting from the ParB locus: DNA signal (left, black line); BsSMC (middle, green); ParB (right, orange) ($n = 215$). **G** Corresponding density plots along the chromosome in all individual cells. Colorbars represent the fold-increase.

chromosome. The data are also in line with the previously reported chromosome rearrangements upon BsSMC deletion or ParB deletion[7,23,24], whereupon chromosomes appeared to lose the second diagonal over time in ensemble Hi-C maps.

## Discussion

Direct microscopy observation of bacterial chromosomes in single cells can reveal novel insights into the underlying organization and dynamics of the DNA within the nucleoid. In *E. coli*, for example, direct observation of chromosomes in expanded-volume cells did resolve the local DNA condensation in the left and right-chromosomal arm as well as a lack of structure within the *ter* domain[15]. The approach furthermore yielded insights into chromosome replication[29,32] and the role of the MukBEF SMC and MatP proteins[14,31] in chromosome organization.

Here, we applied a combination of cell-volume expansion with chromosome-copy control in *Bacillus subtilis* to directly resolve the structure of the chromosome with single-cell microscopy. This revealed that without a cell-boundary constraint, the unconfined shape of the chromosome is a crescent shape. Most of the time, the crescent chromosome was positioned with its convex side near the edge of the spheroidal cells. Our data showed only a weak correlation between the

chromosome size and cell size (Fig. S4), suggesting that cell wall confinement in the spheroidal cells had only a minor effect on the crescent shape. While the crescent was the predominant chromosome phenotype (58%, Fig. S11), other shapes were also observed at smaller fractions (multilobed, compacted) which could arise from multiple chromosomes and not yet fully expanded chromosomes after the cell expansion, respectively. We further found that the crescent shape was stably maintained over long time periods (at least 45 min) while still freely moving in the cell volume (Fig. S15B).

Our data show that the shape of the nonconfined *B. subtilis* chromosome is a crescent shape (Figs. 1, 2). This well compatible with the previous models where Hi-C data suggested a crescent shape or an S-shape[7]. This crescent-shape chromosome is likely deformed when confining it to the rod shape of the bacterium. Previous studies indicated that the chromosome adopts a helicoidal structure in cylindrical cells[6,7]. As the crescent-shape chromosome needs to be tightly compacted within the cylindrical cell shape, introduced helicity might be the most efficient way for the compaction while still regulating origin and terminus regions to be at different cell poles.

The crescent shape may well be the predominant chromosomal shape in bacteria that possess an SMC-ScpAB complex that loads at the origin of replication. Multiple Hi-C studies point toward similar

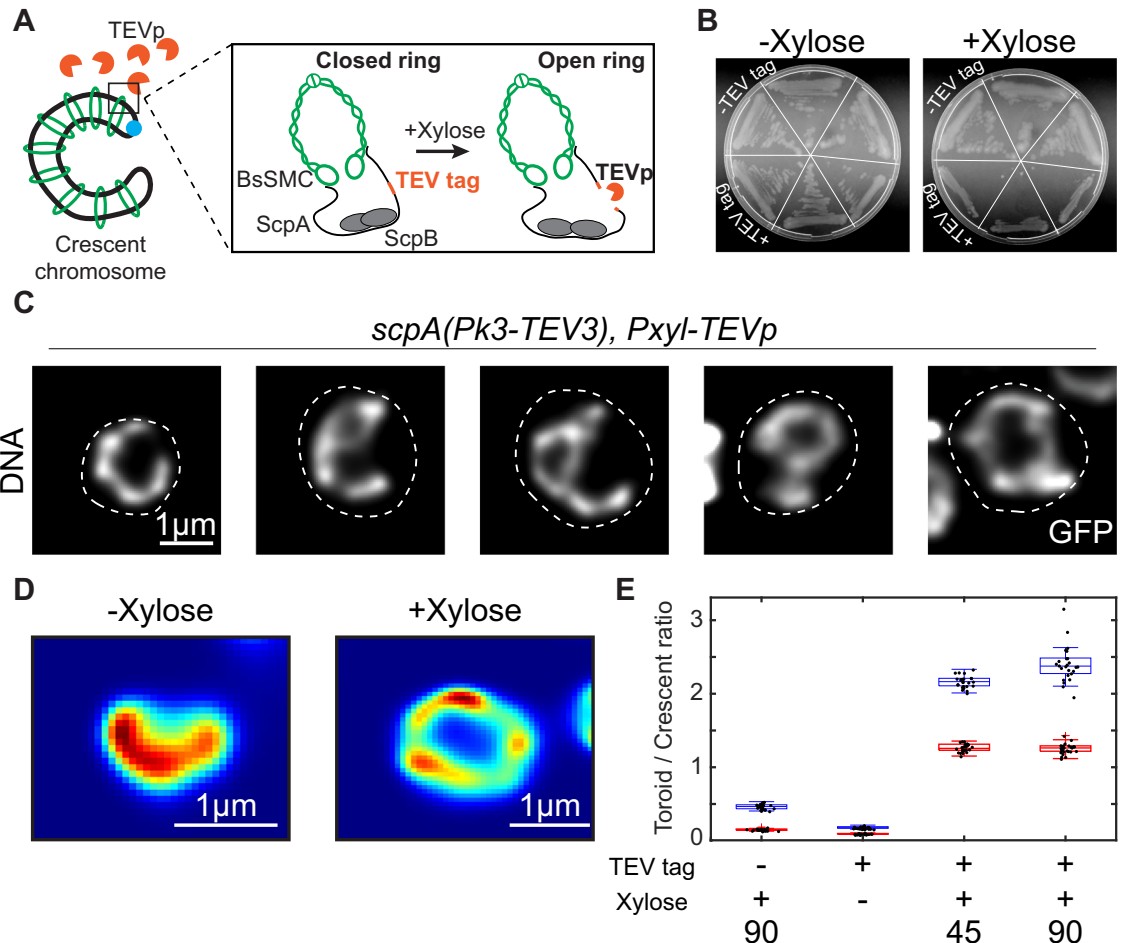

**Fig. 4 | Disruption of BsSMC leads to the opening of the chromosome into a toroidal shape. A** Graphical representation of BsSMC along the crescent-shape chromosome in strain BSG219 (*dnaB(ts-134); hbsu-gfp; scpA(Pk3-tev3); P_{xyl}-TEVp*). Zoomed region details the effect of the TEV protease that cuts the TEV cleavage site within the ScpA protein, which opens the BsSMC ring structure that supposedly is holding together the chromosome arms. **B** Plating assay of strain BSG219 (depicted as +TEV) and its control BSG217 (depicted as -TEV) that is lacking the TEV cleavage site, in the absence (left) and presence (right) of 0.5% Xylose in the agar medium grown at 30 °C. **C** Selected examples of the disrupted chromosomes in strain

BSG219 after 60 min of xylose (0.5%) in the liquid medium at 39 °C. Dashed white line represents the cell outline. The experiment performed in triplicates gave similar results. **D** Representative examples of the crescent and toroidal chromosome shape with (right) and without (left) 0.5% xylose treatment. **E** Relative ratios of toroidal to crescent chromosomes in four different conditions. Each condition was performed using at least three biological replicates. Box plots represent the mean ± SE value after bootstrapping (*n* = 25) per user and per condition. See Supplementary Fig. S16 and Methods for details on these estimates.

chromosomal features in various bacteria possessing a SMC-ScpAB/ParABS systems (e.g., ParB origin localization, a condensed origin, zipped chromosomal arms)[20,24,26–28,66]. By contrast, *E. coli* chromosomes do not adopt these features in Hi-C maps[67], while they also do not possess a ParABS system like most bacteria[68]. Interestingly, we found that acute knock-down of BsSMC resulted in large-scale chromosome reorganization where the left and right-chromosomal arms lost proximity and the chromosome opened into a toroidal shape (Figs. 4, S9 and refs. 7,24), which is very reminiscent of the chromosome shape in widened *E. coli* cells[15]. While the two chromosomal arms thus are mutually connected with BsSMC complexes, one may ask what maintains the compaction of the individual chromosome arms. Recent work in *E. coli* suggests that such a compaction in chromosome interaction domains is largely induced by transcription[65], which we observed in our rifampicin-treated *B. subtilis* cells too (Fig. S18). The stable crescent shape in *B. subtilis* chromosome could thus be maintained by a combination of the arm-zipping action of SMC proteins and the continuous presence of supercoiling arising from the interplay of RNA polymerases and topoisomerases.

The DNA density along the chromosome was found to be very inhomogeneous. Our work expands on previous work by the Nollmann

and Koszul labs who used Hi-C in combination with super-resolution microscopy to resolve the chromosome structure including its underlying domains[7]. They observed that the replication origin contained a dense DNA region that they termed High-Density Region (HDR). Our method, which resolved the full chromosome structure in expanded single cells, also observed an increased DNA density in the origin region at the tip of the crescent-shaped chromosome. In many instances (67% of cells), the condensed DNA cluster close to origin contained even over 40% of the entire genome within it (Fig. S8). The increased density in the origin region may arise from multiple underlying biological processes. First, ParB proteins were proposed to condense the DNA near the *ori* site and form a partition complex by bringing multiple *parS* sites during the initial steps of chromosome segregation[7,50,69,70]. Second, ParB proteins have been proposed to recruit BsSMC proteins to the origin of the chromosome where a juxtaposition between the chromosomal arms is initiated[23,52,54,59]. As our data indeed show an increased BsSMC content near the origin of replication, higher DNA density regions may result from an increased frequency of chromosome folding by BsSMC proteins. Third, the origin-of-replication genomic section contains highly transcribed genes in most bacteria, including *B. subtilis*[71]. These genes are often

accompanied by a high degree of supercoiling and plectoneme formation. This can further increase DNA condensation within the origin region compared to the rest of the chromosome. Maintaining the origin region in a condensed state could have important implications for chromosome segregations via entropic forces[72,73]; because highly condensed regions will be preferentially pushed toward the cell periphery in a cylindrical cell[73]. Accordingly, deletions of BsSMC and ParB proteins mostly result in delayed origin segregation[63,74] and aberrant chromosome segregation in *B. subtilis*, as nascent origins would stick together and reduce segregation fidelity[52,54].

Our data also showed an enrichment of BsSMC at the origin region (Fig. 3). This is in agreement with ChIP-seq data[7,23,24,75] and indicates that only a fraction of the total amount of BsSMC complexes on the chromosome are actively zipping the chromosomal arms, while the largest fraction resides at the origin close to ParB. The enrichment in the origin region might be exacerbated due to SMC:SMC collisions caused by an increased SMC:chromosome ratio due to the replication-halting condition[76,77]. Although a recent biochemical study points to the direct interaction between ParB and BsSMC proteins[59], the precise mechanism of BsSMC recruitment to the origin and its subsequent release to zip the entire chromosome is not yet well resolved. Our observation that higher local amount of SMC proteins resulted in higher DNA content (Fig. S14), hints to the fact that SMC proteins that are locally stalled or blocked still remain on the DNA to locally condense it.

This study would benefit from a comparison between Hi-C maps for untreated rod-shaped bacteria and widened spheroidal bacteria under our imaging conditions. Indeed, a comprehensive study that would include comparative Hi-C mapping, three-dimensional modeling of the chromosome shape from Hi-C data, and fluorescent microscopy of chromosomes under unconfined conditions can be expected to give further detailed insight into the complex chromosome organization of *B. subtilis*.

Our study investigated the *B. subtilis* chromosome organization via direct live-cell imaging. The data revealed the intrinsic crescent chromosome shape and the distribution of ParB and BsSMC proteins within single cells, as well as a disruption of chromosome shape in single cells upon BsSMC knock-down. We observed that the origin of the replication is maintained in a condensed state even under non-confining conditions. The cell-shape manipulation imaging approach can be applied to other bacteria, allowing for single-cell real-time imaging of the chromosome and the dynamics of intercellular processes.

## Methods
### Strain construction
All strains were constructed using the homologous recombination-based cloning described in detail in Diebond-Durant et al.[78]. In brief, we

used a transformation of recombinant DNA to create *B. subtilis* strains at the *smc*, *scpA*, *parB*, *amyE*, *hbsu* and *thrC* loci by allelic replacement harboring natural competence induced by the stationary phase growth as well as starvation due to media depletion (1–2 h incubation in SMM medium lacking amino acid supplements) the same as described in detail Diebold-Durand et al.[78]. We selected the new strains on SMG-agar plates in the presence of the appropriate antibiotic at 30 °C for 16 h. We verified the correct genotype of single colonies by antibiotic resistance profiling on solid agar plates, colony PCR, and locus sequencing and stored the correct clones at −80 °C.

Strains with modified smc alleles were created as described in ref. 78. In *Bacillus subtilis*, the *smc* gene is located in the operon containing *rncS-smc-ftsY* genes. To avoid potential adverse regulatory impacts on *ftsY* gene, the selection marker for *smc* modifications was positioned downstream of the operon, precisely after *ftsY*.

### Bacterial growth conditions
Prior to all experiments, bacterial strains stored at −80 °C, were streaked on a 1×NA (Oxoid) agar plate (1.5% agarose). Plates were incubated at 30 °C for 10−16 h, overnight to obtain single colonies, and used for a maximum of 2 days after that. One day before the experiments we picked single colonies and incubated them in 5 ml liquid osmoprotective medium for 10−16 h, overnight. For all experiments performed, we examined at least three separate individual colonies which represent biological triplicates, and used them under the same procedure. Osmoprotective media (from now also referred to as SMM + MSM) used in this study were modified from Kawai et al. and composed of 2×MSM (40 mM magnesium chloride, 0.2 M−1 M succinate, 40 mM maleic acid, 0.2% yeast extract) mixed in 1:1 relation with 2×SMM (0.04% magnesium sulfate, 0.2% sodium citrate dehydrate, 0.4% ammonium sulfate, 1.2% potassium dihydrogen phosphate, 2.8% dipotassium phosphate) and supplemented with 0.2% L-tryptophan, 1 µg/ml ferric ammonium citrate, 0.1% glutamic acid, 0.02% casamino acids (Bacto™), 1 mg/ml L-threonine. We supplemented the overnight cultures with 5 µg/ml kanamycin, 1 µg/ml erythromycin, 5 µg/ml chloramphenicol, 100 µg/ml spectinomycin where applicable (see Table 1), but the cultures on the day of experiments were not exposed to selection antibiotics. We made 2×MSM solutions with the following range of succinate concentrations 0.2 M, 0.4 M, 0.6 M, 0.8 M, 1 M, for the purpose of testing osmoprotective conditions with the sugar concentration ranging from 100 to 500 mM (see Fig. S1). On the day of the imaging experiments, we diluted the overnight cultures 50× in the total of 10 ml osmoprotective media. We grew the cultures for 3 h (30 °C, 200 rpm orbital shaking), until they reached the early exponential phase, and then induced replication halt either by supplementing with 2 mM IPTG or moving them to 39 °C and 200 rpm orbital shaking, depending on the strain, for a total of 30−180 min depending on the experiment. In the case of TEV protease expression experiments

**Table 1 | Bacterial strains used in this study**

| Name | Genotype | Origin |
|---|---|---|
| BSG001 | 168 ED, trpC2 | Gruber lab |
| BSG217 | 168 ED, dnaB(ts-134; K85E), ΔamyE::Hbsu-GFP::CAT, scpA::specR, thrC::Pxyl-TEVp::ermC, trpC2 | This study |
| BSG219 | 168 ED, dnaB(ts-134; K85E), ΔamyE::Hbsu-GFP::CAT, scpA(Pk3-TEV3)::specR, thrC::Pxyl-TEVp::ermC, trpC2 | This study |
| BSG1001 | 1A700, trpC2 | Gruber lab |
| BSG4595 | 1A700, ParB-mScarlet::kan, amyE::Phyperspank-opt.rbs-sirA (spec), trpC2 | This study |
| BSG4596 | 1A700, amyE::Phyperspank-opt.rbs-sirA (spec), trpC2 | This study |
| BSG4610 | 1A700, parB-mScarlet::kan, amyE::Phyperspank-opt.rbs-sirA (spec), hbsU-mTorquais::CAT, trpC2 | This study |
| BSG4612 | 1A700, smc-mGFPmut1 ftsY::ermB, amyE::Phyperspank-opt.rbs-sirA (spec), trpC2 | This study |
| BSG4623 | 1A700, smc::-mGFP1mut1 ftsY::ermB, hbsU-mTorquais::CAT, ParB-mScarlet::kan, amyE::Phyperspank-opt.rbs-sirA (spec), trpC2 | This study |
| BSG5503 | 1A700, ΔparB::kanR, amyE:Phyperspank-opt.rbs-sirA::specR, trpC2 | This study |
| BSG5522 | 1A700, ParB-mScarlet:kanR, amyE::Phyperspank-opt.rbs-sirA::specR, RTP-GFP::CAT(campbell), trpC2 | This study |

using the strains BSG217 and BSG219 (Table 1), we added 1.5% xylose 45 min before imaging for timecourse imaging (Fig. 4). For the real-time imaging of BsSMC degradation we included the xylose (at 1.5% final concentrations) into the agar pad just before imaging.

## Conversion to L-form spheroid cells

After growing the bacteria for a targeted time of replication halting, we subjected the shaking bacterial cultures to lysozyme treatment. We made a 100 mg/ml fresh stock of lysozyme on the day of the experiment by mixing Lysozyme from chicken egg white (Sigma-Aldrich) with 1×PBS (140 mM NaCl, 10 mM phosphate buffer, and 3 mM KCl, pH 7.4, Phosphate Saline Buffer, (Sigma-Aldrich)) and kept it on ice for the duration of that day (4−6 h). Depending on the temperature of the cell growth we added the lysozyme to a final concentration of 400 μg/ml either 20 min, when grown at 39 °C, or 30 min before, when grown at 30 °C.

## Fluorescence imaging

After cell growth and necessary treatments in liquid culture, we transferred ~3 ml of the liquid cell culture to the 35 mm glass bottom dish (MatTek), which we beforehand passivated with 100 μg/ml UltraPure™ BSA (Invitrogen) for 15−20 min. We specifically avoided using any pipetting steps after the lysozyme treatment to maintain the integrity of spheroidal cells and prevent them from lysis due to shear forces. We then centrifuged the imaging dishes custom-made holders at 400 g for 4 min in a swingout centrifuge with 96-well plate holders (Centrifuge 5430R, Eppendorf). We removed residual supernatant containing non-adherent cells by pipetting at the edge of the dish and placed an agarose pad (~4 × 4 mm) containing 1x osmoprotective medium (lacking yeast extract) and 0.2% low melting agarose (Promega, Madison, USA) on top of the glass bottom. Note that yeast extract needed to be removed from the medium used for making the soft imaging agar pad due to residual background fluorescence in the green channel (488 nm). We placed a small 11 × 11 mm glass coverslip on top of the agar pad to assure the cells were not floating in the medium during imaging due to liquidity of 0.2% agarose and to prevent premature evaporation and drift.

We carried out wide-field Z-scans out using a Nikon Ti-E microscope with a 100× CFI Plan Apo Lambda Oil objective (NA = 1.45). The imaging stage was maintained at 39 °C for experiments using temperature-sensitive mutants, and at room temperature for mutants affected by IPTG expressing of SirA protein. We excited DAPI using SpectraX LED (Lumencor) filter cube with $\lambda_{ex}/ \lambda_{bs}/ \lambda_{em}$ = 363−391/425/435−438 nm, GFP and SYTOXGreen using filter cube with $\lambda_{ex}/ \lambda_{bs}/ \lambda_{em}$ = 485−491 nm/506 nm/501−1100 nm, and mScarlet and STYOXOrange using filter cube with $\lambda_{ex}/ \lambda_{bs}/ \lambda_{em}$ = 540−580/585/592−668 nm. We used an Andor Zyla USB3.0 CMOS Camera to collect the fluorescent signals. We took 9−11 Z-slices of step size l = 200 nm, which accounts for a total of 1.8−2.2 μm of scanning volume, that we fed into the deconvolution software (see below).

## Solid agar plating assays

For testing the growth differences in presence/absence of xylose in the strains carrying *scpA(Pk3-TEV3)::specR* locus (Fig. 4B), we used a standard LB-agar (1.5% agarose). As described above, we grew the strains on separate plates carrying the selective antibiotic (here 5 μg/ml chloramphenicol and 100 μg/ml spectinomycin) one day before the experiment. We picked three single colonies from each plate and re-streaked them in a radial orientation on the plate containing LB-agar or LB-agar + 1.5% xylose. The plates were left to incubate at 30 °C for 16 h, after which we imaged them on a Gel imager (BIO-RAD Chemidoc™).

## Growth curves

For monitoring bulk growth (Fig. S6), we first grew the strains in the selective media overnight, as mentioned previously. On the day of the experiment, we diluted the overnight cultures 40x in the fresh SMM + MSM medium and grew them until the OD reached ~0.2−0.3. We then again diluted the cultures to the final OD of 0.01 and distributed in 96-well plate (Nunc), with the final volume of 170 μl. The plates were loaded into an Infinite 200Pro fluorescence plate reader (Tecan, Männedorf, Switzerland) and incubated at 30 °C with the orbital shaking (2.5 mm amplitude) for a period of 24−48 h. Cell density was measured at 600 nm at 15 min intervals. We performed all experiments using at least biological triplicates, and in some instances more.

## Quantitative RT-PCR on genomic DNA

To verify the replication halt by expressible *sirA* gene, we performed qPCR as shown by Stokke et al.[79] using [PrimeTime™, IDT]. We grew the bacterial cultures BSG4610 and BSG4595 under the same conditions and media mentioned above. We grew bacterial samples in biological triplicates for both treated and untreated sample (with or without IPTG). We induced the replication halt by adding 2 mM IPTG for 120 min. Following this incubation, all samples were centrifuged (13,000 × g, 2 min) to collect cells and remove the surrounding medium. Next, we extracted the genomic DNA using [PureLink™ Genomic DNA kit, Invitrogen] following the standard protocol for gram positive bacteria supplied with the kit.

To perform qPCR we used the premixed primer pairs that bind within the *ori* (Fw:CGAGCTGGTTCCTACAATATCA, Rev: ATGACGGCGGACAATCAA, with the qPCR probe: FAM 520, ZEN/Iowa Black™ AGAGAGCGCTTGAAGCAGTAAAGCA) and *ter* region (Fw: GAACCTAATGTTAAGGAAGAAAGCC, Rev: GGACCAGTTAGGGCGATATTT, with the qPCR probe: FAM 520, ZEN/Iowa Black™ CCTAGCAATGATGTCGACACTGATGGA) of the *B. subtilis* chromosome. The PrimeTime™ Gene Expression Master Mix, primers, and qPCR probes were thawed on ice for 10 min. We then briefly vortexed the reagents and centrifuged them to collect all liquid at the bottom of the tube. Primers and probes were ordered premixed, and upon arrival we spun the mixtures at 750 × g for 30 s. We dissolved these mixtures in 100 μl of TE buffer to obtain 20x solutions. We mixed the solutions in water (final concentration of 9 mM of each Fw and Rev primer and 2.5 mM of the qPCR probe), together with the PrimeTime™ Gene Expression Master Mix (1×), as well as 3 ng of each of genomic DNA sample for the final volume of 20 μl for the reaction. Mixtures of primer pairs and probes for *ori* and *ter* were used in separate reactions. We initially set the sample to 95 °C for 3 min. Following this step, we set the amplification for 40 cycles (denaturation 95 °C for 15 s, annealing and extension 60 °C for 60 s). We obtained the primer efficiency for both *ori*-primer and *ter*-primer pairs at >91%. The *ori:ter* ratios were determined per individual technical triplicate sample (for each of biological triplicates) and represented in Figs. S7, S12.

## Deconvolution microscopy

For deconvolution microscopy we used Huygens Professional deconvolution software (Scientific Volume Imaging, Hilversum, The Netherlands), using an iterative Classic Maximum Likelihood Estimate (CMLE) algorithm. We measured the point spread function (PSF) experimentally by using 200 nm Tetraspeck beads (Invitrogen) and the recommended Huygens Professional guidelines (https://svi.nl/Point-Spread-Function-(PSF)). Stacks of 9−11 Z-slices were fed into the Huygens Professional software and we deconvolved each signal channel separately.

## Structured illumination microscopy imaging

For SIM microscopy, samples were prepared as previously described, except using two rounds of centrifugation with 3 ml cell culture in order to obtain higher cell density to accommodate for the smaller camera FOV. We used a Nikon Ti-E microscope with an AiRSIM module and a 100× CFI Apo Oil objective (NA = 1.49). We used a 3D-SIM option with the 5 pos x 3 angles for imaging our live samples, and a Z-stack of

**Table 2 | Oufti detection parameters used in this study**

| Parameter name | Value |
|---|---|
| **cellDetection** | |
| ThreshFactorM | 0.9925 |
| ThreshMinLevel | 0.8 |
| EdgeSigmaL | 1 |
| **spotDetection** | |
| minHeight | 0.002 |
| minWidth | 0.5 |
| maxWidth | 10 |
| Adjusted squared error | 0.4 |
| **objectDetection** | |
| Bkg subtraction method | 3 |
| Bkg subtraction threshold | 0.1 |
| Bkg filter size | 8 |
| Smoothing range (pixels) | 3 |
| LOG filter | 0.1 |
| Sigma of PSF | 1.5 |
| Fraction of object in cell | 0.4 |
| Minimum object area | 50 |

11 positions with $l = 200$ nm in between. NIS-Elements (version 5.2.1) software was used for image reconstruction where we carefully adhered to the Nikon N-SIM reconstruction guidelines and parameters to avoid reconstruction artifacts (Fig. S5).

## Image processing and analysis

For obtaining the cell outlines and counting fluorescent foci within each cell, as well as for measuring distances between ParB-mScarlet and BsSMC-GFP foci within each cell, we used the image segmentation and analysis software Oufti[80]. We obtained subpixel precision *B. sub-tilis* cell outlines using the cellDetection tool in Oufti with the parameters shown in Table 2. We used the detectSignal tool for fluorescent foci detection of the ParB-mScarlet and BsSMC-GFP signals using the selection parameters shown in Table 2.

We used custom-written MATLAB scripts, modified from the original publication of Kaljevic et al.[81], to extract the fluorescent foci positions, count the number of foci per cell, and determine the distances between foci in each channel. These scripts are open-source and available online (https://zenodo.org/records/7733856).

For a systematic and detailed analysis of the chromosome shape, shown in e.g. Figs. 2 and 3, we used home-developed Matlab code based on earlier work from Wu et al.[15]. The image processing covered the following elements:

**Selection.** First, we used Fiji/ImageJ to user-select ROIs around round cells based on the phase image alone, yielding typically ~$10^2$ cells per field-of-view (FOV). This ROI list was imported in Matlab. For each ROI, we obtained a mask outlining the cell wall from the wide-field image. Within this cell mask, we analyzed the signals of the other color labels, depicting the patterns representing DNA, SMC, and ParB.

**Screening.** For a randomly chosen FOV, the selected round cells displayed a large variety of patterns and artifacts and sometimes included misdetections due to the thresholding. The latter were screened out by a simple cell roundness criterion. Furthermore, multiple ParB spots could lie in different focal planes such that only one was observed per focal plane. To make sure we selected single chromosomes, we applied a projection of maximum intensity along the defocus stack and accepted only those cells that had a single ParB spot appearing in maximum projection.

**Backbone pattern analysis.** Here, we made use of the characteristic shape of the chromosome pattern. As seen in-plane, the patterns formed crescent arcs that broadly followed the inside cell wall over a broad annular section of 90°–180°. As described before, such an arc-like structure could well be described in an annular coordinate system, where a 'backbone' center line runs over the chromosome arc[15]. Using this backbone as a length axis, we sampled the relative intensity of the respective labels per unit annular section along this length. To compare many cells, we performed this sampling over a fixed, representative length, starting at the angle where the ParB signal maximizes, since this maximum represents the location of the *ori* site on the chromosome. We thus created intensity vs distance profiles for all labels. Notably, the cells with the crescent-shape chromosome may appear in a regular fashion or lie upside down. Therefore, starting from the *ori* location, the crescent may point clockwise or counterclockwise. To compare all cells, we flipped the profiles such that for all, the crescent intensity moved toward positive length values. We collected all these profiles in demographs shown in Fig. 2I and Fig. 3G. Each line in a demograph represents a heatmap of the DNA density profile from the origin (position of 0 µm) to the end of the chromosome. These lines are stacked for all cells (y-axis) to make the common trends well visible.

**Foci and cluster analysis.** Following an earlier described method[15], we described the measured intensity patterns as a sum of clusters. Here, the chromosome pattern was described as a group of separate clusters. Each of these clusters was then described with a sub-group of psf-limited spots. For this sub-group, any of the spots was within one psf of at least one other. This resulted in a smooth, compact shape of this cluster. On the other hand, all spots in another cluster would be further away than one psf, so that there was always a distinguishable gap between each two neighboring clusters. In this way, the sometimes complicated 'blob patterns' can be described in a compact way as a limited set (typically, 4–5 per cell) of optically separable clusters, with each of these clusters coming with a particular xy position and a relative intensity (as compared with the intensity counted in the entire cell area). In the case of a compact focus, the cluster description simplifies in just one psf-sized spot. If a single cluster extended beyond one psf, such as the somewhat smeared-out spots observed for the ParB label, this approach still provided us with an intuitive description of the amount of fluorescence associated with one ParB spot. Finally, the xy positions of the cluster analysis can be correlated to the above-described backbone length axis, allowing to readily and quantitatively compare the relative intensities of mutually associated clusters of DNA, ParB and SMC (Fig. 3).

## Chromosome shape selection

We were unsuccessful in obtaining the ParB-mScarlet tag in strains BSG217 and BSG219, containing HbsU-tag, $P_{xyl}$-*TEV*p and/or ScpA-TEV3 in multiple cloning attempts. While we cannot distinguish between single and partially replicated chromosomes, due to strain construction issues, we attempted to quantitatively distinguish between torus-shaped chromosomes and crescent-shaped chromosomes in the presence of 0.5% xylose. For this, we performed the following manual user selection. We collected one image from four different samples (Figs. 4E, S16), which describe four experiments with different expected outcomes. Each of these was intensity thresholded to detect ROIs with separated DNA patterns based only on the continuous fluorescent signal. Next, a randomly picked ROI, containing a signal from a single cell, was presented to a user for classification of this pattern. The user could choose from one of four categories (toroid, crescent, compact, other/undefined) without knowing from which original image the ROI was taken. This was done for a series of 100 picks each, with no ROIs being presented to the user more than once within this series. To obtain an estimate for sampling variation, we repeated this series 20

times per user for a total of 2000 selected ROIs. To evaluate possible user bias, the procedure was done independently by two users (MT and JK) shown in Fig. S16. We then traced back the described ROIs (in one of those four categories) to the original images and plotted the selection distribution shown in Figs. 4E and S16.

## Chromosomal loci counting – *ori* and *ter*

To quantify the *ori:ter* ratio (Fig. S7) and ensure that the replication-halted cells had a single chromosome, we imaged the strain BSG5522 under the same conditions as previously mentioned. We localized the cell outlines and positions of *ori* and *ter* loci using Oufti[80] and the parameters shown in Table 2. We performed additional filtering to account for the low signal-to-noise ratio in the *ter* signal channel (TRP-GFP), caused by the presence of only a few proteins at *Ter* sites. We removed the detected spots that contained less than 15% of the total signal present in the cell, using custom-written MATLAB software (see *Data and materials availability*).

## Visualization and representation

All fluorescent and phase images were visualized in the open-source image-analysis software Fiji[82]. We used a calculated microscopic pixel size of 65.35 nm/px in all images for scale bar creation. Adobe Illustrator 2020 was used for visualization and manuscript figures. After reconstructing the SIM microscopy images, we used SIMcheck software package for visualization[83] (Fig. S5) and a Fiji in-built function "3D project" to create a 3D reconstruction of the crescent chromosome (Movie S1), as well as the plug-in 3D viewer to create the isosurface plots of the same image (Movie S2). The same was used to visualize the deconvolved 3D stacks in Movies S3, S4.

## Reporting summary

Further information on research design is available in the Nature Portfolio Reporting Summary linked to this article.

## Data availability

All data included in the manuscript are available in the main text or the supplementary material. All raw data from plots, generated in this study have been deposited in the Zenodo database under accession code https://zenodo.org/records/7733856.

## Code availability

All code, analysis scripts, including the test dataset for new users, used in this study have been deposited in the Zenodo database under accession code https://zenodo.org/records/7733856.

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

## Acknowledgements

We thank Jaco van der Torre for useful discussions and advice during the
project. We acknowledge funding support for the work in C.D. lab by the
European Research Council Advanced Grant 883684 as well as by
project OCENW.GROOT.2019.012 which is financed by the Dutch
Research Council (NWO). We also acknowledge funding for the work in
S.G. lab by the Swiss National Science Foundation (grant number:
310030_197770).

## Author contributions

Conceptualization: M.T., S.G., C.D. Cloning and strain construction: M.T.,
F.P.B., H.A. Experimental work: M.T. Formal analysis: M.T., A.J., J.K.
Methodology: M.T., F.P.B., J.K. Visualization: M.T. Funding acquisition:
S.G., C.D. Supervision: S.G., C.D. Writing – original draft: M.T., C.D.
Writing – review & editing: M.T., F.P.B., J.K., H.A., A.J., S.G., C.D.

## Competing interests

The authors declare no competing interests.

## Additional information

**Supplementary information** The online version contains
supplementary material available at

Cees Dekker.

**Peer review information** *Nature Communications* thanks the anon-
ymous reviewer(s) for their contribution to the peer review of this work. A
peer review file is available.

