## [Peer Review File · Nature Communications]

Direct observation of a crescent-shape chromosome in *Bacillus subtilis*REVIEWER COMMENTS

Reviewer #1 (Remarks to the Author):

In this study, Tisma et al., use a combination of high-resolution microscopy and cell-shape manipulation to investigate the shape of the Bacillus chromosome. They first expanded the cell volume via treatment of cells with lysozyme. Further, they controlled the chromosome copy number to ensure these cells had only a single chromosome. Using this system, they visualized the Bacillus chromosome with DAPI/ Sytox/ Bacillus HU. They found that the chromosome took up a crescent shape, with maximum DNA density around the origin region (as assessed by the localization of ParB). This crescent shape transitioned to a toroid upon disruption of the SMC complex, suggesting that the two chromosome arms are 'zipped' together by the SMC proteins. These observations are in contrast to the work reported by the same group with regards to E. coli chromosome organization, where the chromosome already adopts a toroidal shape in wild type conditions. The authors attribute the difference in chromosome structure to the distinct mechanisms of chromosome segregation in E. coli vs B. subtilis.

Overall, this is a well-executed study that provides important insights into Bacillus chromosome organization. Of note are the results providing direct evidence for chromosome arm zipping by SMC proteins (previously inferred from ensemble Hi-C experiments). The experiments are elegantly designed, the writing and interpretation of the data are clear and well-supported by experimental evidence. I have a few comments the authors should address to strengthen their conclusions:

Major comments:

1. Authors should provide flow cytometry controls to show that the cells in their experiments indeed have only a single chromosome
2. Could the authors comment on the dynamics of the crescent-shape chromosome? Is the ori region stationary, or does it move throughout the cell volume? Does the crescent shape also display dynamic motion in the cells?
3. Authors conclude that a large fraction of the genome is located within the 'origin cluster'. This is inferred from imaging the chromosome via SyG staining and localization of fluorescently-tagged ParB. Authors should provide confirmation of the same with analysis of chromosome labelled with HbsU (as used in Fig. 3D).
4. Authors conclude that the chromosome transitions from crescent to toroid shape upon disruption of the BsSMC ring. Authors should provide data visualizing the localization of the left and right arms of the chromosome (via use of the array system or orthogonal parS-ParB chromosome labelling system) in crescent and toroid shaped chromosomes in support of this conclusion.
5. Does ParB remain localized in cells where SMC has been disrupted? Do these cells still have a majority of their chromosome located within the 'origin cluster'?
6. In the discussion section, authors should comment on why the crescent-shaped chromosome appears towards the edge of the cell.

Minor comments:

1. In the introduction, authors should also discuss Hi-C evidence from chromosome arm zipping in other bacteria (for example Caulobacter).
2. Page 3: results: What is meant by "to accommodate the change"?

3. Page 4: first paragraph. Authors should check the figure referencing as referred figures are not accurate for the conclusions drawn. For example, Fig. 1D is used to conclude on the average volume increase in cells after lysozyme treatment.
4. Page 9: first paragraph. “resolve the actual structure” could be modified to “resolve the actual chromosome structure”
5. Pag 10: first paragraph. This paragraph needs significant rewriting to coherently bring out the comparison between *E. coli* and *B. subtilis*. In its present form, the text is rather confusing.
6. Pag 10: first paragraph. What do authors mean by “aberrative segregation”.

Reviewer #2 (Remarks to the Author):

This manuscript uses optical microscopy methods to gain insight into the 3D organization of the chromosome in *Bacillus subtilis* cells.

The main findings are that *B. subtilis* chromosome displays a crescent shape with a non-uniform DNA density, with increased density near the replication origin. In addition, SMC and ParB are shown to localize primarily close to this same region. Disruption of SMC leads to the opening of the crescent shape and created a rather torus conformation.

Novelty issues

Previous studies using super-resolution methods as well as Hi-C established that the BS chromosome folds with the two arms pairing each other, that the distribution of density in the chromosome is inhomogeneous (this gave rise to high-density domains or HdR) (Marbouty, 2015). Earlier studies had also shown that ParB and SMC localize mainly to the *oriC* region (Gruber, 2009; Sullivan, 2009). SMC depletion was also shown to disrupt the pairing of the chromosome (Wang, 2015; Marbouty, 2015) and lead to a torus like conformation (see graphical abstract of Marbouty, 2015) . The Hi-C study shown earlier even proposed a crescent-like shape with a higher density near *oriC* (Marbouty, 2015). Finally, this study also showed that a toroidal conformation appears in absence of DNA replication, and that crescent-like structures appear as replication is allowed to proceed.

Thus, I find that many of the findings in this ms seem to have been previously reported using similar (e.g. 3D-SIM on live cells) or complementary methods (e.g. Hi-C). I feel it is a missed opportunity not to have focused more on the single cell analysis of 3D conformations (which previous reports could not analyze) and relate these to the localization of proteins involved (eg. SMC/ParB) in these single 3D conformations (see also below).

The methods in this manuscript were previously used by the authors in *E. coli*: conversion of rod-like cells into spheroidal-like cells; or tracing of the intensity of fluorescence along the *E. coli* chromosome to report regions of higher DNA densities and changes in overall chromosome conformations. If there are differences in the approaches used, then it would have been useful to highlight them.

Major issues

It is unclear whether the conformation of the chromosome - in conditions where it is not any more constrained by the cell wall - are relevant at all to the conformation of the chromosome in the cell. Indirectly or directly the cell wall imparts structure to the nucleoid and regulates its segregation and the localization of main genetic loci (like the origin or the terminus). Thus, it is unclear if the structures shown are relevant to wild-type rod-like cells. The authors should at least discuss these points or provide a rationale for why they think these conformations are relevant.

It is correct that HiC models are derived from population data. Thus, an ensemble chromosome conformation displaying a crescent shape, and a transition to a toroidal shape in absence of SMC, do not mean that the chromosome will be adopting these conformations in single cells. I was thus hoping to see a study where the ability of microscopy to image chromosome conformations in single-cells would be used to: characterize if crescent/toroidal shapes could be seen in single nucleoids, and map the variation in single-nucleoid conformations. An attempt at this analysis is provided in Fig S3, but frustratingly this analysis seems to be based (unless I am confused) only on wide field imaging where the spatial resolution is rather limited (as shown in Fig 1E). Why did the authors not use SRM images?

From this analysis (Fig. S3) authors conclude that only 40% of cells show the intrinsic chromosome shape. What do the other nucleoids look like? Why none of these other conformations (which represent the majority) are not displayed? Does this not mean that in most cells the chromosome does NOT display a crescent like shape? How are these distributions affected by the lack of cell wall confinement, cell wall interactions mediated by protein factors? A prominent presentation, and a discussion of the variations in conformations would be very interesting to our understanding of how bacterial chromosomes are really organized beyond models derived from ensembles.

Also, it would have been important to link these variations in 3D DNA conformations to changes in the distributions of SMC/ParB. Unfortunately, these analysis are not provided. Histograms of localizations for DNA/SMC/ParB are shown in different plots, and it is unclear whether there is any relation between them.

While the authors display single nucleoid conformations, is it unclear how many cells displayed this kind of fluorescence distribution.

Authors explain that they observe an increase in DNA fluorescence proximal to the ParB focus. However, Figs. 2H-I show that often cells display maxima of DNA intensity that do not seem to be at zero (where ParB is). This does not seem to match their statement. Is there an explanation for this?

Smaller issues

Many citations are wrong (the wrong journal/year are cited, or page/year information are missing).

Number of biological replicates are not reported, or I did not find them in the legends.

Surprising to see low number of cells, often close to 1000 per experiment, while current super-

resolution microscopes can image thousands of bacterial cells per field of view.

Reviewer #3 (Remarks to the Author):

In this article, Tisma et al demonstrate that L-forms of *B. subtilis* cells organize their chromosome in a crescent-shape. Using a cool technique to break the SMC packaging of the chromosome, the authors show that the crescent shaped chromosome is maintained specifically by SMC. My major concern with this work resides with the authors' conclusion that untreated *B. subtilis* cells form the crescent-shaped chromosomes. More controls are needed to support that interpretation. Below I list specific suggestion for additional experiments, most of which are control experiments.

Major

1. The evidence of *B. subtilis* organizing its chromosome in a crescent shape is limited to one technique that includes a drastic treatment. I am concerned that the lysozyme treatment could cause the chromosome to form the crescent shape. Can Hi-C be performed with the lysozyme-treated cells to correlate patterns observed with Hi-C of untreated *Bacillus* cells? If not, can the authors think of other ways to confirm that the crescent shape is independent of treatment? Maybe test another monoderm with the same technique.
2. The claim that in 67% of cells, >40% of the entire genome was clustered within 500nm of oriC is very interesting. This needs to be confirmed. For instance, strains with fluorescently labeled chromosomal loci away from ori (i.e., using tetO-TetR arrays) can be used to show that indeed they localize near ori.
3. Figure 2. A quantification of ori:ter ratio is needed to confirm that replication has not partially initiated in *sirA* mutant cells. Replication could have initiated and stopped soon after. However, ParB fluorescently labeled would not be able to separate multiple oriCs if the replication halted without segregation.
4. Figure 3. *parB* knockout as a control strain is needed to confirm that SMC is responsible for the crescent shape.
5. The graphical representation in Figure 4A suggest that the terminus is located at the other tip of the crescent shape. Labeling the terminus would help to visualize the ori and ter loci at the poles of the presumably crescent shape.
6. The discussion is weak. The only things discussed are that more condensed DNA are found near ori and that SMC impacts chromosome shape, which have all been shown before. The interesting finding of this work is the possibility of a crescent shaped chromosome. The discussion should be expanded to address questions like, why would this shape form? what would drive it? what is filling the space along the crescent inner curvature? Are there any other examples of this type of chromosomal organization, perhaps in eukaryotic cells? Why (referring to structure of cell) would this crescent shape be formed by *B. subtilis* but not by *E. coli*? etc.

Minor:

1. Were experiments performed in independent triplicates? This should be explicitly mentioned in the figure legend.
2. Were samples blinded for image analyses?
3. All the quantification plots need error bars (i.e., Figures 2C, 2E, 4E...)

4. Figure 1AB seem to not be mentioned in the text.
5. The Supplemental figures are mentioned on the text not in a numerical order. For instance, S4 and S6 are mentioned before S3 and S5, respectively.
6. Include reasoning behind using *ftsY::ermB* mutation in the background of strains used.
7. Figure S2. Were different temperatures tested? The data do not seem to include temperature, but the strain is *dnaB134ts*
8. In discussion, briefly discuss why only subpopulation of cells from Fig. S3B form chromosomes in a crescentus shaped
9. "Using previously stablished analyses, we detected DNA clusters....". Include a brief explanation to the method.
10. "Instead, DNA segregation in *B. subtilis* is mainly driven by the ParABS system". My understanding was that the ParABS system required during sporulation that is why the system is not essential in *B. subtilis*.
11. "We observed that SMC proteins positioned as a combination of typically 1-2 fluorescent foci ... (Fig. 3B)": include a reference if this is how SMC typically localizes in *B. subtilis*.
12. "which did not affect growth in cylindrical cells (Fig. 1C, S1C)". Figures 1C and S1C do not include growth data
13. "In contrast to the observations in *E. coli*, the final chromosome size...". Briefly explain those observations for those not familiar with the *E. coli* work.

Reply to the Reviewers

(original comments in black font; our reply in blue font)

Reviewer #1

In this study, Tisma et al., use a combination of high-resolution microscopy and cell-shape manipulation to investigate the shape of the *Bacillus* chromosome. They first expanded the cell volume via treatment of cells with lysozyme. Further, they controlled the chromosome copy number to ensure these cells had only a single chromosome. Using this system, they visualized the *Bacillus* chromosome with DAPI/ Sytox/ Bacillus HU. They found that the chromosome took up a crescent shape, with maximum DNA density around the origin region (as assessed by the localization of ParB). This crescent shape transitioned to a toroid upon disruption of the SMC complex, suggesting that the two chromosome arms are ‘zipped’ together by the SMC proteins. These observations are in contrast to the work reported by the same group with regards to *E. coli* chromosome organization, where the chromosome already adopts a toroidal shape in wild type conditions. The authors attribute the difference in chromosome structure to the distinct mechanisms of chromosome segregation in *E. coli* vs *B. subtilis*.

Overall, this is a well-executed study that provides important insights into *Bacillus* chromosome organization. Of note are the results providing direct evidence for chromosome arm zipping by SMC proteins (previously inferred from ensemble Hi-C experiments). The experiments are elegantly designed, the writing and interpretation of the data are clear and well-supported by experimental evidence. I have a few comments the authors should address to strengthen their conclusions:

We thank the Reviewer for the very positive appraisal.

Major comments:

1. Authors should provide flow cytometry controls to show that the cells in their experiments indeed have only a single chromosome.

We thank the Reviewer for this suggestion and we accordingly attempted these experiments. *Bacillus subtilis* has, however, the tendency to grow in chains of cells that prevent the faithful analysis of DNA content for single cells by flow cytometry. Cell chaining is less pronounced in a medium supporting slow growth as used here for single cell imaging. We thus attempted to perform flow cytometry using SYTO9 staining. However, as seen below (Fig. R1), the DNA content signal still shows a broad distribution, which likely results of cell chaining. Under replication halting conditions, the DNA profile did not change drastically: although an increase in lower DNA content cells can be seen, the flow cytometry data are not clear enough to be discriminative.

Figure R1. Flow cytometry measurements of the replication halt in BSG217 and BSG219.

2. Could the authors comment on the dynamics of the crescent-shape chromosome? Is the ori region stationary, or does it move throughout the cell volume? Does the crescent shape also display dynamic motion in the cells?

The chromosome with its origin moves freely throughout the cell volume over time, albeit slowly. We measured the chromosome dynamic movement as now shown in Fig. S9. We performed additional timelapse experiments for the origin locus, tracking with time frames of 1s. The origin displayed dynamic movements within 5-6 pixels (~300nm) on the time scale of ~10s. Our analysis shows that the FWHM of the origin movement distribution (FWHM = 89 nm) is only slightly larger than the cell itself (FWHM = 56 nm). Thus, we conclude that there is no substantial motion of the ori within the chromosome on the 10 second time scale. Additional discussion on the new results is included in the main text of the paper and Fig. S9.

3. Authors conclude that a large fraction of the genome is located within the ‘origin cluster’. This is inferred from imaging the chromosome via SyG staining and localization of fluorescently-tagged ParB. Authors should provide confirmation of the same with analysis of chromosome labelled with HbsU (as used in Fig. 3D).

We thank the Reviewer for this suggestion. To address this, we successfully constructed a strain for Hbsu imaging: BSG4610 (Genotype: *1A700*, *ParB-mScarlet::kan*, *amyE::Phyperspank-opt.rbs-sirA (spec)*, *hbsU-mTurquoise2::CAT*, *trpC2*) although the strain showed slower growth

under the same medium conditions possibly due to tagging on the native hbsU locus. We imaged these cells under the same conditions. The data showed the same crescent chromosomal structure (Fig. S11), and the origin cluster analysis showed similar condensation levels as in our initial SytoxGreen analysis (Fig. 2F-J). This is now included in Fig. S11.

4. Authors conclude that the chromosome transitions from crescent to toroid shape upon disruption of the BsSMC ring. Authors should provide data visualizing the localization of the left and right arms of the chromosome (via use of the array system or orthogonal parS-ParB chromosome labelling system) in crescent and toroid shaped chromosomes in support of this conclusion.

We thank the reviewer for this suggestion. We attempted to construct such strains but were unsuccessful with incorporating the orthogonal ParB_{PMT1}/parS_{PMT1} (or ParB_{P1}/parS_{P1}) system into our *B. subtilis* strain BSG219 (which contains TEV-tag kleisin and HU-GFP) in four different attempts. However, to test whether the chromosome transitions from crescent to toroid shape upon loss of normal SMC distribution, we constructed a Δ parB strain which does not have the SMC loader (ParB protein). This strain shows almost exclusively toroidal chromosomes with chromosomal arms separated.

This is now described in more detail in the main text, as requested by Reviewer #3, and the data on the toroidal chromosome shape in the absence of ParB is now included in the new Fig. S16.

5. Does ParB remain localized in cells where SMC has been disrupted? Do these cells still have a majority of their chromosome located within the ‘origin cluster’?

We thank the Reviewer for this suggestion. Aiming to address this, we created the following strains for ParB imaging:

1. 168 ED, dnaB(ts-134; K85E), Δ amyE::Hbsu-GFP::CAT, scpA::specR, thrC::Pxyl-TEVp::ermC, parB-mScarlet::kanR, trpC

2. 168 ED, dnaB(ts-134; K85E), Δ amyE::Hbsu-GFP::CAT, scpA(Pk3-TEV3)::specR, thrC::Pxyl-TEVp::ermC, parB-mScarlet::kanR, trpC2

However, we were ultimately unsuccessful to yield useful cell-imaging data with these. Even though the strains showed growth on plates, they exhibited a very sick phenotype under the microscope (i.e., elongated cells, altered DNA distribution throughout the cells, spontaneous bulging of cells) and the cells could not confidently be used for data acquisition.

Hence, we addressed this question with additional data analyses. We analyzed the DNA distribution over the donut chromosomes and observed that a steady local cluster persists in most cells – see Fig. R2. The analysis done was the same as for crescent chromosomes (see Methods).

Figure R2: Fluorescent intensity along the chromosome after SMC disruption. A) DNA intensity along the contour of the chromosome. Black line shows the average normalized intensity obtained from all cells. Gray lines display arbitrarily chosen individual examples. The position of the largest DNA cluster is set at the 0 μm position. B) DNA density along the chromosome in all individual cells. Color bar represents the fold-increase. Cells are ordered from top to bottom in terms of contrast. C) Representative examples of “open” chromosomes after SMC disruption; Top – fluorescent Hbsu-GFP signal; Bottom – brightfield image. Scale bar – 1 μm .

6. In the discussion section, authors should comment on why the crescent-shaped chromosome appears towards the edge of the cell.

We thank the Reviewer for this suggestion. This is now included in the discussion.

Minor comments:

1. In the introduction, authors should also discuss Hi-C evidence from chromosome arm zipping in other bacteria (for example *Caulobacter*).

We thank the Reviewer for this suggestion. A brief description of data from other bacteria is now included in the introduction, citing relevant work in *Caulobacter*, *Corynebacterium*, *Agrobacterium*, *Pseudomonas*.

2. Page 3: results: What is meant by “to accommodate the change”?

This was meant to reflect the “fast conversion to spheroidal cells” from the previous sentence. For clarity, we have changed this now.

3. Page 4: first paragraph. Authors should check the figure referencing as referred figures are not accurate for the conclusions drawn. For example, Fig. 1D is used to conclude on the average volume increase in cells after lysozyme treatment.

We apologize for this mistake. We thank the Reviewer for pointing this out. We have corrected the figure referencing and changed the phrasing appropriately.

4. Page 9: first paragraph. “resolve the actual structure” could be modified to “resolve the actual chromosome structure”

We included this suggestion by the Reviewer.

5. Pag 10: first paragraph. This paragraph needs significant rewriting to coherently bring out the comparison between E. coli and B. subtilis. In its present form, the text is rather confusing.

The discussion has now been significantly altered following the comments from the Reviewers (and in this rewriting, this particular paragraph has almost entirely been omitted).

6. Pag 10: first paragraph. What do authors mean by “aberrative segregation”.

This paragraph has been altered on the suggestion of Reviewer #3 and the expression is no longer there.

Reviewer #2

This manuscript uses optical microscopy methods to gain insight into the 3D organization of the chromosome in Bacillus subtilis cells.

The main findings are that B. subtilis chromosome displays a crescent shape with a non-uniform DNA density, with increased density near the replication origin. In addition, SMC and ParB are shown to localize primarily close to this same region. Disruption of SMC leads to the opening of the crescent shape and created a rather torus conformation

We thank the Reviewer for carefully examining our work, for this brief summary and for the comments that we address below.

Novelty issue:

Previous studies using super-resolution methods as well as Hi-C established that the BS chromosome folds with the two arms pairing each other, that the distribution of density in the chromosome is inhomogeneous (this gave rise to high-density domains or HdR) (Marbouty, 2015). Earlier studies had also shown that ParB and SMC localize mainly to the oriC region (Gruber, 2009; Sullivan, 2009). SMC depletion was also shown to disrupt the pairing of the chromosome (Wang, 2015; Marbouty, 2015) and lead to a torus like conformation (see graphical abstract of Marbouty, 2015). The Hi-C study shown earlier even proposed a crescent-like shape with a higher density near oriC (Marbouty, 2015). Finally, this study also showed that a toroidal conformation appears in absence of DNA replication, and that crescent-like structures appear as replication is allowed to proceed.

Thus, I find that many of the findings in this ms seem to have been previously reported using similar (e.g. 3D-SIM on live cells) or complementary methods (e.g. Hi-C). I feel it is a missed

opportunity not to have focused more on the single cell analysis of 3D conformations (which previous reports could not analyze) and relate these to the localization of proteins involved (eg. SMC/ParB) in these single 3D conformations (see also below).

The Reviewer is correct in noting that several insights coming from this paper are well in line with previously suggested chromosomal structures. Here we were able to resolve these structures at the level of single live cells of relaxed shapes, which was never done before and which yielded a number of new findings.

Furthermore, some notes on the valuable earlier studies may be phrased: Hi-C maps come from ensemble studies that have the intrinsic disadvantage of cell fixation, which potentially might result in chromosome compaction, which could convolute the conclusions. The before-mentioned beautiful imaging study (Marbouty et al., 2015. Mol Cell) showed that the chromosome adopts a helicoidal structure (i.e. not a crescent shape), similar to a previous study (Berlitzky et al., 2008. PNAS). Such a structure likely is the result of cell confinement rather than reflecting its intrinsic chromosome shape, as we do not observe such structures in our shape-relaxed cells. The toroidal shape upon replication halt appeared as a result of SMC accumulation and collision from multiple parS sites (Brandão et al., 2022. NSMB).

In our discussion, we now elaborate on the novelty of our work.

The methods in this manuscript were previously used by the authors in *E. coli*: conversion of rod-like cells into spheroidal-like cells; or tracing of the intensity of fluorescence along the *E. coli* chromosome to report regions of higher DNA densities and changes in overall chromosome conformations. If there are differences in the approaches used, then it would have been useful to highlight them.

While the overall effect (a single chromosome within an expansion cell volume) was the same, the methodology that was used to achieve this was quite different between *E. coli* and *B. subtilis*. The previous *E. coli* study from our lab used an antibiotic treatment (A22) and standard medium in order to grow *E. coli* cells larger into a pancake/lemon shape. Here, however, we induced a complete cell wall removal (i.e. not partial by blocking MreB via A22), and we conserved the cells in a spherical/L-form shape in an isosmotic/slightly hypoosmotic medium (300-500mM osmolyte). As this treatment is much faster (5-20min vs 2-3h for A22 in *E. coli*), we expect lower perturbations of the chromosome shape and size. The Methods section explains in detail how the cells were constructed and we have added a short reference to *Kawai et al.* on this point in the main text.

Major issues:

It is unclear whether the conformation of the chromosome - in conditions where it is not any more constrained by the cell wall - are relevant at all to the conformation of the chromosome in the cell. Indirectly or directly the cell wall imparts structure to the nucleoid and regulates its segregation and the localization of main genetic loci (like the origin or the terminus). Thus, it is unclear if the structures shown are relevant to wild-type rod-like cells. The authors should at least discuss these points or provide a rationale for why they think these conformations are relevant.

In our opinion, an understanding of the spatial structure of the bacterial structure starts with resolving its intrinsic shape. We agree that the cell wall likely imparts structure to the nucleoid – which is of interest in itself. Yet, that does not invalidate the interest of understanding the intrinsic shape of an unconfined bacterial chromosome.

We added this notion to the discussion.

It is correct that HiC models are derived from population data. Thus, an ensemble chromosome conformation displaying a crescent shape, and a transition to a toroidal shape in absence of SMC, do not mean that the chromosome will be adopting these conformations in single cells. I was thus hoping to see a study where the ability of microscopy to image chromosome conformations in single-cells would be used to: characterize if crescent/toroidal shapes could be seen in single nucleoids, and map the variation in single-nucleoid conformations. An attempt at this analysis is provided in Fig S3, but frustratingly this analysis seems to be based (unless I am confused) only on wide field imaging where the spatial resolution is rather limited (as shown in Fig 1E). Why did the authors not use SRM images?

We do agree with the Reviewer on the interest of microscopy to image chromosome conformations in single cells. The analysis in Fig. S3B represents the analysis of BSG217 strain which has HU-GFP as a chromosome label as well as a thermosensitive replication halting system (via *dnaB-ts*). The drawback of this strain is that it does not strictly ensure the presence of only a single chromosome per cell. For this reason, we focused the in-depth analysis of cell-to-cell variations to a strain that has both ParB label (ensuring a single chromosome in that cell) and HU/DNA label – as reported in Fig.2. To avoid overinterpretation of our data (e.g. by including cells that have more than 1 chromosome, cf. Fig. S3B), we thus based our main conclusions on cells from the strain from Fig. 2, which were pre-selected to contain a single chromosome (see Methods). Here we reported the DNA density, chromosome size, and position within the spherical cells, as well as cell-to-cell variations in the condensation levels (see Fig. 2H-J).

From this analysis (Fig. S3) authors conclude that only 40% of cells show the intrinsic chromosome shape. What do the other nucleoids look like? Why none of these other conformations (which represent the majority) are not displayed? Does this not mean that in most cells the chromosome does NOT display a crescent like shape? How are these distributions affected by the lack of cell wall confinement, cell wall interactions mediated by protein factors? A prominent presentation, and a discussion of the variations in conformations would be very interesting to our understanding of how bacterial chromosomes are really organized beyond models derived from ensembles.

We thank the Reviewer for this request for clarification. To provide such clarity, we have now expanded the figure to show all shapes were observed. Notably, the crescent shape is the most dominant shape, while highly globular condensed and bi-/tri-lobed shapes showed a lower occurrence (see revised Fig. S3).

In these experiments, we used a replication halt time that was shorter than twice the doubling time, in order to avoid conditions that caused a loss of the second diagonal in HiC data (Brandão et al., 2022. NSMB). Additionally, we applied rigorous selection criteria to ensure that only *single*

chromosomes were analyzed (see Methods). We thus characterized the overall shape of cells within these populations. From experience with strain BSG4595 (with a labelled ParB), we observed that bi- and tri-lobed nucleoids often possessed multiple ParB foci. Those are discarded in our analysis in Figure 2, but likely still present in Figure S3B-C.

Also, it would have been important to link these variations in 3D DNA conformations to changes in the distributions of SMC/ParB. Unfortunately, these analysis are not provided. Histograms of localizations for DNA/SMC/ParB are shown in different plots, and it is unclear whether there is any relation between them.

We thank the Reviewer for this excellent suggestion. We now built correlation plots between these three quantities – ParB, SMC, and DNA, which are now included in Fig. S13.

While the authors display single nucleoid conformations, is it unclear how many cells displayed this kind of fluorescence distribution.

In fact, the majority of cells displayed this kind of distribution. Specifically, the fluorescence distribution in different strains (e.g., BSG4595 or BSG4623) are displayed in Figures 2I and 3G. Figures 2H and 3F show that on average there is an increase of DNA amount in the origin region (close to ParB) and a decay towards the other end of the crescent chromosome, while individual fluorescence distributions for each cell are shown in the demographs below them (Figures 2I and 3G).

Authors explain that they observe an increase in DNA fluorescence proximal to the ParB focus. However, Figs. 2H-I show that often cells display maxima of DNA intensity that do not seem to be at zero (where ParB is). This does not seem to match their statement. Is there an explanation for this?

Figure 2J shows the distribution of the DNA clusters along the crescent chromosome, starting from the 0-position (= the position of ParB focus). Black dots represent the main cluster (with most DNA in it) and the grey dots represent secondary clusters. Looking at the black data, most clusters are focused near the 0-position. More precisely, taking only $\pm 250\text{nm}$ (~ 4 pixels) around the zero, we see that $\sim 70\%$ of the clusters localizes near the ParB positions.

The data such as in Fig. 2H indeed shows some peaks away from the 0 position, but even in these examples, the main peak is closer to 0-position – see Fig. R3. Fig. 2I shows the distribution for many cells, and depicts that most of the curves have their main peak near ParB.

Figure R3. DNA intensity along the contour of the crescent chromosome adapted from Figure 2H. Black line shows the average normalized intensity obtained from all cells ($N=619$). Gray lines display arbitrarily chosen individual examples. The red line in the panel on the right shows a highlighted example (red arrow in left panel) with a prominent secondary condensation peak that is distant to the origin at $0 \mu\text{m}$.

Smaller issues:

Many citations are wrong (the wrong journal/year are cited, or page/year information are missing).

We thank the Reviewer for pointing this out and we apologize for this oversight. After moving from a new reference manager, our citations changed formats and we ended up like this. This is now corrected.

Number of biological replicates are not reported, or I did not find them in the legends.

We added this information to the figure captions and in methods (in the “Bacterial growth conditions” paragraph).

Surprising to see low number of cells, often close to 1000 per experiment, while current super-resolution microscopes can image thousands of bacterial cells per field of view.

While 1000 is not a low number, we note that the statistics were often lowered by the stringent selection for single chromosomes. Figure R4 shows an example of such selection. For each cell, we measure the relative fluorescent content of the main ParB focus. When the relative intensity dropped below 60% (dotted line in left panel), it is likely that the signal was divided across more than one focus and the cell was rejected. In the right panel, it can be seen that a large fraction of cells was rejected based on this criterion.

Figure R4. Single chromosome selection criteria. A) Relative fluorescent signal within the main *ParB* cluster. Red dashed line represents a threshold of 60% total fluorescence signal for selecting single-origin cells. B) Cell selection count before (blue) and after (red) single origin screening.

Reviewer #3

In this article, Tisma et al demonstrate that L-forms of *B. subtilis* cells organize their chromosome in a crescent-shape. Using a cool technique to break the SMC packaging of the chromosome, the authors show that the crescentus shaped chromosome is maintained specifically by SMC. My major concern with this work resides with the authors' conclusion that untreated *B. subtilis* cells form the crescent-shaped chromosomes. More controls are needed to support that interpretation. Below I list specific suggestion for additional experiments, most of which are control experiments.

We thank the Reviewer for appreciating our approach and for the suggested controls which we address below.

Major:

1. The evidence of *B. subtilis* organizing its chromosome in a shape is limited to one technique that includes a drastic treatment. I am concerned that the lysozyme treatment could cause the chromosome to form the crescentus shape. Can Hi-C be performed with the lysozyme-treated cells to correlate patterns observed with Hi-C of untreated *Bacillus* cells? If not, can the authors think of other ways to confirm that the crescentus shape is independent of treatment? Maybe test another monoderm with the same technique.

We thank the Reviewer for these suggestions. Hi-C experiments were difficult for us to perform for technical/logistics reasons because we lost the person that was doing Hi-C at the Gruber lab. We agree that doing Hi-C experiments on expanded and rod-shaped cells could yield interesting results, however it would require an extensive study to infer to the internal 3D chromosomal shape from a contact plot (i.e., Hi-C experiments, modelling of the Hi-C maps, mutant strains).

Based on the comments of this Reviewer and Reviewer #2, we have now expanded the discussion section to comment on the possible effects of cell wall and cell wall removal on the chromosome shape. Notably, we observe that the chromosomes adopt a crescent shape immediately after L-form cells reshape (<1-3 min), while they maintain their crescent shape for long times (>45 min, Fig. S14). Imaging of BSG219 (in presence of Xylose) and BSG5503 strain, showed, by contrast,

that chromosomes do not adopt a crescent shape (but rather a toroidal shape), even though they underwent the same experimental treatment. Unfortunately, extensive cloning for chromosome labeling and replication halting in another monoderm bacteria is beyond the scope of the work.

2. The claim that in 67% of cells, >40% of the entire genome was clustered within 500nm of oriC is very interesting. This needs to be confirmed. For instance, strains with fluorescently labeled chromosomal loci away from ori (i.e., using tetO-TetR arrays) can be used to show that indeed they localize near ori.

We thank the reviewer for this suggestion to confirm our quantitative claim for origin condensation. The same was pointed out by Reviewer 1 (comment 3). We now provided confirmation of this using our newly constructed strain BSG4610 (Genotype: *IA700, ParB-mScarlet::kan, amyE::Phyerspank-opt.rbs-sirA (spec), hbsU-mTorquais::CAT, trpC2*), which is now included in Fig. S10. We imaged these cells under the same conditions. The data showed the same crescent chromosomal structure (Fig. S11), and the origin cluster analysis showed similar condensation levels as in our initial SytoxGreen analysis (Fig. 2F-J). In this strain we see 63% and 65% (two replicates) of the cells having more than 40% of their genome within the origin cluster.

3. Figure 2. A quantification of ori:ter ratio is needed to confirm that replication has not partially initiated in *sirA* mutant cells. Replication could have initiated and stopped soon after. However, ParB fluorescently labeled would not be able to separate multiple oriCs if the replication halted without segregation.

We thank the reviewer for suggesting this important control. For this purpose, we now constructed a strain containing origin label and terminus label: *BSG5522: IA700, ParB-mScarlet:kanR, amyE::Phyerspank-opt.rbs-sirA::specR, RTP-GFP::CAT (campbell), trpC2*. The strain showed replication halt in an almost identical fashion as previously mentioned strain BSG4596 (Figure 2B-E), and we did not observe multiple termini with single unresolved origins. This is now shown in Figure S7, as supporting information for our conclusions in Figure 2.

Please note that in the presence of SMC and ParA, origins will often segregate uninterruptedly, and both of these proteins and their corresponding genes are unaffected in our strain BSG4595.

4. Figure 3. *parB* knockout as a control strain is needed to confirm that SMC is responsible for the crescent shape.

We thank the reviewer for suggesting this control to show that both of the key factors (ParB and SMC) affect the crescent shape observed under standard conditions.

For this purpose, we constructed a strain BSG5503 (*IA700, ΔparB::kanR, amyE:Phyerspank-opt.rbs-sirA::specR, trpC2*). Here we observed the chromosome shapes entirely disrupted, showing either a toroidal shape (as seen in *ScpA* knock-down, Fig. 4) or entirely undefined chromosome shapes. This is now shown in Fig. S16, and correspondingly mentioned in the main text.

5. The graphical representation in Figure 4A suggest that the terminus is located at the other tip of

the crescentus shape. Labeling the terminus would help to visualize the ori and ter loci at the poles of the presumably crescentus shape.

The Figure 4A did not specifically indicate the location of the terminus in our crescent-shaped chromosomes. However, we agree with the reviewer that it is helpful and important to visualize the terminus of the replication along the crescent chromosome.

Accordingly, we used a Ter-labeled strain BSG5522 under our cell-expansion conditions, and observed that in the presence of a single origin, the terminus indeed located at the opposite end of the crescent chromosome. This is now included in the Fig. S10 and mentioned in the main text.

6. The discussion is weak. The only things discussed are that more condensed DNA are found near ori and that SMC impacts chromosome shape, which have all been shown before. The interesting finding of this work is the possibility of a crescentus shaped chromosome. The discussion should be expanded to address questions like, why would this shape form? what would drive it? what is filling the space along the crescentus inner curvature? Are there any other examples of this type of chromosomal organization, perhaps in eukaryotic cells? Why (referring to structure of cell) would this crescentus shape be formed by *B. subtilis* but not by *E. coli*? etc.

We have expanded the discussion section to include the questions recommended by the Reviewer. The discussion section is now fully restructured to cover the points by this and other Reviewers.

Minor:

1. Were experiments performed in independent triplicates? This should be explicitly mentioned in the figure legend.

We thank the Reviewer for this question. We now mentioned this in figure legends and in Methods in the “bacterial growth conditions” paragraph.

2. Were samples blinded for image analyses?

The analysis was indeed blinded, in two different ways:

In Figures 1-3, all cells were selected only on the basis of the brightfield image, i.e. without examining the signals from either ParB or DNA. We chose to select all isolated cells within the field of view (i.e. those that did not tightly touch other cells whereupon signals would overlap and convolute). All following analyses were performed based on this initial selection, and hence the user could not choose particular cells of interest.

In Figure 4 (as is explained in detail in the Methods), all samples from all conditions were pooled together and single cells were shown to two independent users in a randomized way and with no information of the sample or visibility of nearby cells. In this way, we avoided the user seeing the entirety of the sample and thus getting biased to which condition the cells came from.

3. All the quantification plots need error bars (i.e., Figures 2C, 2E, 4E...)

We thank the reviewer for this suggestion. Error bars were now added to all bar plots and histograms where appropriate, and the representation of error bars was added to the figure legends.

4. Figure 1AB seem to not be mentioned in the text.

We thank the Reviewer for pointing this out. These figure panels are now referenced properly and in order.

5. The Supplemental figures are mentioned on the text not in a numerical order. For instance, S4 and S6 are mentioned before S3 and S5, respectively.

These particular figures are now mentioned in order.

6. Include reasoning behind using *ftsY::ermB* mutation in the background of strains used.

We thank the Reviewer for this suggestion. This is now included in the Methods section under “Strain construction”.

7. Figure S2. Were different temperatures tested? The data do not seem to include temperature, but the strain is *dnaB134ts*

We specified the growth conditions under “bacterial growth conditions” in the Methods section. Strains were always grown at 30°C initially, while the replication was halted by moving to 37°C. Growth temperatures for *dnaB134ts* strains are now additionally, included in the Fig.1 and Fig.4 legends.

8. In discussion, briefly discuss why only subpopulation of cells from Fig. S3B form chromosomes in a crescentus shaped

We thank the reviewer for this suggestion. This in now included in the discussion section.

9. “Using previously stablished analyses, we detected DNA clusters....”. Include a brief explanation to the method.

We now briefly expanded this in the main text, and referred to the extensive explanation in the Methods section under “Image processing and analysis” subsection.

10. “Instead, DNA segregation in *B. subtilis* is mainly driven by the ParABS system”. My understanding was that the ParABS system required during sporulation that is why the system is not essential in *B. subtilis*.

Initially the roles of ParA and ParB were discovered in sporulation. However, many follow up studies have shown that they are heavily involved in various other chromosome-organization roles (most notable replication initiation and chromosome segregation) that are independent of their role

in sporulation. While the system indeed is not essential in *Bacillus subtilis* (and other Gram+ bacteria), mutations or deletion of the system or its individual components severely impact the chromosome dynamics. Defects include elongated cells, overreplication, aberrant origin separation, impaired SMC loading, and increased formation of anucleate cells. For a more general overview of the functions of the system and effects of perturbing individual components, we now added a review on the *ParABS* system: Kawalek et al. *Microorganisms*. 2020 Jan 11;8(1):0. “Rules and Exceptions: The Role of Chromosomal ParB in DNA Segregation and Other Cellular Processes”. doi: 10.3390/microorganisms8010105. PMID: 31940850.

11. “We observed that SMC proteins positioned as a combination of typically 1-2 fluorescent foci ... (Fig. 3B)”: include a reference if this is how SMC typically localizes in *B. subtilis*.

We now included two citations of the initial SMC localization in rod cells [Gruber et al. 2009, Sullivan et al 2009].

12. “which did not affect growth in cylindrical cells (Fig. 1C, S1C)”. Figures 1C and S1C do not include growth data

We thank the Reviewer for this correction. For clarity, we changed our statement from “growth” to “phenotype”.

13. “In contrast to the observations in *E. coli*, the final chromosome size...”. Briefly explain those observations for those not familiar with the *E. coli* work.

We thank the Reviewer for the suggestion. We have rephrased this part to briefly introduce the main takeaway from previously published *E. coli* results and cited the work accordingly.

REVIEWER COMMENTS

Reviewer #1 (Remarks to the Author):

My comments have been addressed satisfactorily.

Reviewer #2 (Remarks to the Author):

I appreciate the efforts made by the authors to answer the many requests of the reviewers, which provide additional support for their claims. I still have a few points that were not addressed in their rebuttal.

Original Question:

It is unclear whether the conformation of the chromosome - in conditions where it is not any more constrained by the cell wall - are relevant at all to the conformation of the chromosome in the cell. Indirectly or directly the cell wall imparts structure to the nucleoid and regulates its segregation and the localization of main genetic loci (like the origin or the terminus). Thus, it is unclear if the structures shown are relevant to wild-type rod-like cells. The authors should at least discuss these points or provide a rationale for why they think these conformations are relevant.

Answer:

In our opinion, an understanding of the spatial structure of the bacterial structure starts with resolving its intrinsic shape. We agree that the cell wall likely imparts structure to the nucleoid – which is of interest in itself. Yet, that does not invalidate the interest of understanding the intrinsic shape of an unconfined bacterial chromosome.

We added this notion to the discussion.

Comment:

The concept of intrinsic shape is difficult to grasp. The chromosome is a polymer. Its conformation will be dictated by polymer physics, by confinement, and by the action of proteins (e.g. SMC, RNAP, topoisomerases, translocases). Thus, the authors claim to describe the shape of the chromosome in absence of one of these factors: confinement. However, it is unclear to me how the removal of a confinement, and the action of the membrane, influence the action of other proteins affecting chromosome conformation.

Second, the discussion starts with the statement that “the intrinsic shape of the chromosome is a crescent shape”. However, this is in contrast to the 60-70% of cells that DO NOT SHOW this shape.

The crescent shape may exist, but their current data does not support this main conclusion in the Discussion.

Finally, the discussion does not shed light into why a crescent-like conformation may arise. What may maintain the chromosome in this restrained shape? By what mechanisms?

Original Question:

It is correct that HiC models are derived from population data. Thus, an ensemble chromosome conformation displaying a crescent shape, and a transition to a toroidal shape in absence of SMC, do not mean that the chromosome will be adopting these conformations in single cells. I was thus hoping to see a study where the ability of microscopy to image chromosome conformations in single-cells would be used to: characterize if crescent/toroidal shapes could be seen in single nucleoids, and map the variation in single-nucleoid conformations. An attempt at this analysis is provided in Fig S3, but frustratingly this analysis seems to be based (unless I am confused) only on wide field imaging where the spatial resolution is rather limited (as shown in Fig 1E). Why did the authors not use SRM images?

Answer:

We do agree with the Reviewer on the interest of microscopy to image chromosome conformations in single cells. The analysis in Fig. S3B represents the analysis of BSG217 strain which has HU-GFP as a chromosome label as well as a thermosensitive replication halting system (via dnaB-ts). The drawback of this strain is that it does not strictly ensure the presence of only a single chromosome per cell. For this reason, we focused the in-depth analysis of cell-to-cell variations to a strain that has both ParB label (ensuring a single chromosome in that cell) and HU/DNA label – as reported in Fig. 2. To avoid overinterpretation of our data (e.g. by including cells that have more than 1 chromosome, cf. Fig. S3B), we thus based our main conclusions on cells from the strain from Fig. 2, which were pre-selected to contain a single chromosome (see Methods). Here we reported the DNA density, chromosome size, and position within the spherical cells, as well as cell-to-cell variations in the condensation levels (see Fig. 2H-J).

Comment:

- Not clear to me why ParB label ensures a single chromosome? Absence of a ParB label means there are more than one? I am confused by the explanations provided.
- It is unclear to me whether the authors based their analysis in widefield and not on super-resolution. This is not answered either.
- What are the conclusions from these analyses? Most cells (~60-70%) do not display a crescent-like structure. Why is this the case? what is the biological significance of this result? Are crescent-like structures related to specific processes (e.g. initiation of replicore pairing) or cell cycle timing ?

Original Question:

From this analysis (Fig. S3) authors conclude that only 40% of cells show the intrinsic chromosome shape. What do the other nucleoids look like? Why none of these other conformations (which represent the majority) are not displayed? Does this not mean that in most cells the chromosome does NOT display a crescent like shape? How are these distributions affected by the lack of cell wall confinement, cell wall interactions mediated by protein factors? A prominent presentation, and a discussion of the variations in conformations would be very interesting to our understanding of how bacterial chromosomes are really organized beyond models derived from ensembles.

Answer:

We thank the Reviewer for this request for clarification. To provide such clarity, we have now expanded the figure to show all shapes were observed. Notably, the crescent shape is the most dominant shape, while highly globular condensed and bi-/tri-lobed shapes showed a lower occurrence (see revised Fig. S3).

Comment:

Analysis should be performed to provide mechanistic insight into these conformations (see above) otherwise the study remains descriptive.

Original Question:

Also, it would have been important to link these variations in 3D DNA conformations to changes in the distributions of SMC/ParB. Unfortunately, these analysis are not provided. Histograms of localizations for DNA/SMC/ParB are shown in different plots, and it is unclear whether there is any relation between them.

Answer:

We thank the Reviewer for this excellent suggestion. We now built correlation plots between these three quantities – ParB, SMC, and DNA, which are now included in Fig. S13.

Comment:

I appreciate these new analyses with interesting results: (1) ParB does not seem correlated to DNA percentage. The percentage of DNA seems rather stable (~30-30%) however the percentage of ParB varies wildly from 0 to 90% (caption says ParB intensity; which of the two (%/intensity) is it?). This is a very surprising result, however no rationale/explanation of how this fits in existing models is provided in the text. (2) In contrast, bsSMC is correlated to DNA percentage. How do the authors interpret this result? The authors argue in the discussion that multi-lobed/ other non-crescent-like conformations may be related to incompletely replicated chromosomes, but this seems to contradict their filtering approach that searches for cells with single ParB complexes to find 'single chromosomes'. Am I missing something?

Original Question:

The claim that in 67% of cells, >40% of the entire genome was clustered within 500nm of oriC is very interesting. This needs to be confirmed. For instance, strains with fluorescently labeled chromosomal loci away from ori (i.e., using tetO-TetR arrays) can be used to show that indeed they localize near ori.

Answer:

We thank the reviewer for this suggestion to confirm our quantitative claim for origin condensation. The same was pointed out by Reviewer 1 (comment 3). We now provided confirmation of this using our newly constructed strain BSG4610 (Genotype: *1A700, ParB-***mScarlet::**kan**,* *amyE**::**Phyperspank-opt.rbs-sirA* *(spec),* *hbsU-mTorquais**::CAT, trpC2*), which is now included in Fig. S10. We imaged these cells under the same conditions. The data showed the same crescent chromosomal structure (Fig. S11), and the origin cluster analysis showed similar condensation levels as in our initial SytoxGreen analysis (Fig. 2F-J). In this strain we see 63% and 65% (two replicates) of the cells having more than 40% of their genome withing the origin cluster.

Comment:

How do they know that these cells are not replicating? Selection of a single ParB locus does not ensure lack of replication. If these cells started replicating but did not start replicore separation then their estimate of the % of genomic DNA in the origin cluster would be off.

Reviewer #3 (Remarks to the Author):

This re-submission has been improved and most of my comments have been addressed, except for the major one.

Major:

Hi-C is the ideal technique to confirm the authors' exciting findings that Bacillus cells may organize their chromosome in a crescent shape. The results are exciting but they need to be confirmed by a second method, especially when the method is available (Hi-C). The reasoning provided by the authors why they cannot perform this experiment is remarkably weak ("lost the person that was doing Hi-C").

Minor:

1. Figure S10B. Please include a population analyses, quantification. What percent of cells display the ori and ter at the ends as displayed in S10B? 100%?
2. Given that flow cytometry is not feasible to confirm that cells only have one chromosome, did the authors consider performing simple qPCR for ori-to-ter ratio analyses? This method will work regardless of cells forming chains.

Point-by-Point reply to the editors and reviewers comments

In black – reviewers' comments

In blue – authors' responses

Reviewer #1

My comments have been addressed satisfactorily.

We thank this reviewer for the useful comments that improved our manuscript.

Reviewer #2

I appreciate the efforts made by the authors to answer the many requests of the reviewers, which provide additional support for their claims. I still have a few points that were not addressed in their rebuttal.

1. Original Question:

It is unclear whether the conformation of the chromosome - in conditions where it is not any more constrained by the cell wall - are relevant at all to the conformation of the chromosome in the cell. Indirectly or directly the cell wall imparts structure to the nucleoid and regulates its segregation and the localization of main genetic loci (like the origin or the terminus). Thus, it is unclear if the structures shown are relevant to wild-type rod-like cells. The authors should at least discuss these points or provide a rationale for why they think these conformations are relevant.

Answer:

In our opinion, an understanding of the spatial structure of the bacterial structure starts with resolving its intrinsic shape. We agree that the cell wall likely imparts structure to the nucleoid – which is of interest in itself. Yet, that does not invalidate the interest of understanding the intrinsic shape of an unconfined bacterial chromosome. We added this notion to the discussion.

Comment:

The concept of intrinsic shape is difficult to grasp. The chromosome is a polymer. Its conformation will be dictated by polymer physics, by confinement, and by the action of proteins (e.g. SMC, RNAP, topoisomerases, translocases). Thus, the authors claim to describe the shape of the chromosome in absence of one of these factors: confinement. However, it is unclear to me how the removal of a confinement, and the action of the membrane, influence the action of other proteins affecting chromosome conformation.

The reviewer is correct in stating that the chromosome conformation will be dictated by DNA polymer physics, by confinement, and by the action of proteins. In our experiments, we relaxed one of these elements (i.e. the confinement), which yields an interesting result, a crescent shape – which aligns well with findings from HiC techniques.

Following the editorial suggestions and the reviewer comments, we now avoid the language of the intrinsic shape and use instead the 'unconfined chromosome'.

About the effects of (non)confinement on proteins: The removal of the confinement does not seem to affect major genome-folding proteins such as the SMC proteins, as our crescent structure matches previously simulated coarse-grained *B. subtilis* chromosomes (Marbouty *et al. Mol Cell* 2015) and it is the removal of

these proteins that largely affects chromosome structure (see Fig. 4, Fig. S15, S16, S17). The effect of RNAP on the chromosome is mentioned below in the experiments using rifampicin (and new Fig. S18).

Second, the discussion starts with the statement that “the intrinsic shape of the chromosome is a crescent shape”. However, this is in contrast to the 60-70% of cells that DO NOT SHOW this shape. The crescent shape may exist, but their current data does not support this main conclusion in the Discussion.

Following the editorial suggestion and the reviewer comments we rephrased this to mention that this is the most frequent shape that we find in the cells where we could resolve the shape.

Finally, the discussion does not shed light into why a crescent-like conformation may arise. What may maintain the chromosome in this restrained shape? By what mechanisms?

Upon removal of ParB (the loader of SMC proteins) or knocking down SMC proteins, the chromosome loses its crescent shape. We therefore conclude that the crescent chromosome is retained by SMC proteins – see Figures 4, S15, S16, S17. The mechanism of SMC proteins maintaining the crescent shape is proposed to be loop extrusion, which was shown in vitro in eukaryotic SMCs (condensin, cohesin, SMC5/6), and proposed in vivo for bacterial SMCs (see Wang et al. *Science* 2017).

SMC action and the possible role of transcription in maintaining the crescent shape is mentioned in the 4th paragraph of the Discussion section.

2. Original Question:

It is correct that HiC models are derived from population data. Thus, an ensemble chromosome conformation displaying a crescent shape, and a transition to a toroidal shape in absence of SMC, do not mean that the chromosome will be adopting these conformations in single cells. I was thus hoping to see a study where the ability of microscopy to image chromosome conformations in single-cells would be used to: characterize if crescent/toroidal shapes could be seen in single nucleoids, and map the variation in single-nucleoid conformations. An attempt at this analysis is provided in Fig S3, but frustratingly this analysis seems to be based (unless I am confused) only on wide field imaging where the spatial resolution is rather limited (as shown in Fig 1E). Why did the authors not use SRM images?

Answer:

We do agree with the Reviewer on the interest of microscopy to image chromosome conformations in single cells. The analysis in Fig. S3B represents the analysis of BSG217 strain which has HU-GFP as a chromosome label as well as a thermosensitive replication halting system (via dnaB-ts). The drawback of this strain is that it does not strictly ensures the presence of only a single chromosome per cell. For this reason, we focused the in-depth analysis of cell-to-cell variations to a strain that has both ParB label (ensuring a single chromosome in that cell) and HU/DNA label – as reported in Fig.2. To avoid overinterpretation of our data (e.g. by including cells that have more than 1 chromosome, cf. Fig. S3B), we thus based our main conclusions on cells from the strain from Fig. 2, which were pre-selected to contain a single chromosome (see Methods). Here we reported the DNA density, chromosome size, and position within the spherical cells, as well as cell-to-cell variations in the condensation levels (see Fig. 2H-J).

Comment:

- Not clear to me why ParB label ensures a single chromosome? Absence of a ParB label means there are more than one? I am confused by the explanations provided.

Since each chromosomes has one ParB cluster, the ParB label serves as a monitor that enables us to *select for cells that have a single chromosome* in our analysis: Cells with one ParB cluster have one chromosome, whereas cells with two ParB clusters have more than one chromosome. Single chromosomes are achieved by the expression of SirA protein, as extensively described throughout the main text and methods section-

We edited the main text to clarify the characterization further. Furthermore, we refer to the new qPCR quantification of our strains under replication-halting conditions requested by the reviewers and editors.

- It is unclear to me whether the authors based their analysis in widefield and not on super-resolution. This is not answered either.

All analysis is based on super-resolution now in all figures (except Fig. S3). In Fig. S3, it is based on widefield images as this image serves to show that we observed crescent shapes even under widefield microscopy conditions before selection, thus showing that this is not an artifact of superresolution deconvolution. This is now also specifically mentioned in Main text.

- What are the conclusions from these analyses? Most cells (~60-70%) do not display a crescent-like structure. Why is this the case? what is the biological significance of this result? Are crescent-like structures related to specific processes (e.g. initiation of replicore pairing) or cell cycle timing ?

Most multi-lobed chromosomes showed *multiple* ParB-foci when analyzed in the strain BSG4595. This means that there were multiple chromosome origins inside cells. Notably, these cells were *not* included on the shape analysis for Fig. 2.

Most cells with a single chromosome showed the crescent shape (~60%), which is analyzed in Fig. 2 extensively, and now mentioned in Fig. S11. Crescent-like structures are not related to replicore pairing, as we halted initiation of replication in this strain (Fig. S6 and S7). Instead, crescent structures can be attributed to the action of SMC proteins and transcription.

3. Original Question:

From this analysis (Fig. S3) authors conclude that only 40% of cells show the intrinsic chromosome shape. What do the other nucleoids look like? Why none of these other conformations (which represent the majority) are not displayed? Does this not mean that in most cells the chromosome does NOT display a crescent like shape? How are these distributions affected by the lack of cell wall confinement, cell wall interactions mediated by protein factors? A prominent presentation, and a discussion of the variations in conformations would be very interesting to our understanding of how bacterial chromosomes are really organized beyond models derived from ensembles.

Answer:

We thank the Reviewer for this request for clarification. To provide such clarity, we have now expanded the figure to show all shapes were observed. Notably, the crescent shape is the most dominant shape, while highly globular condensed and bi-/tri-lobed shapes showed a lower occurrence (see revised Fig. S3).

Comment:

Analysis should be performed to provide mechanistic insight into these conformations (see above) otherwise the study remains descriptive.

On this point, we respectfully feel that the reviewer is asking too much, since as of yet, no one has obtained a full mechanistic understanding of chromosome organization. Indeed, our study describes, for the first time, the chromosome shape in unconfined *Bacillus subtilis*, and it discusses the likely maintenance of the crescent chromosome.

We added one additional piece of information: The chromosome compaction is partly maintained by transcription, see new Fig. S18. Upon transcription halt, the chromosome loses its crescent shape and forms dispersed and compacted shapes.

4. Original Question:

Also, it would have been important to link these variations in 3D DNA conformations to changes in the distributions of SMC/ParB. Unfortunately, these analysis are not provided. Histograms of localizations for DNA/SMC/ParB are shown in different plots, and it is unclear whether there is any relation between them.

Answer:

We thank the Reviewer for this excellent suggestion. We now built correlation plots between these three quantities – ParB, SMC, and DNA, which are now included in Fig. S13.

Comment:

I appreciate these new analyses with interesting results:

(1) ParB does not seem correlated to DNA percentage. The percentage of DNA seems rather stable (~30-30%) however the percentage of ParB varies wildly from 0 to 90% (caption says ParB intensity; which of the two (%/intensity) is it?). This is a very surprising result, however no rationale/explanation of how this fits in existing models is provided in the text.

(2) In contrast, bsSMC is correlated to DNA percentage. How do the authors interpret this result? The authors argue in the discussion that multi-lobed/ other non-crescent-like conformations may be related to incompletely replicated chromosomes, but this seems to contradict their filtering approach that searches for cells with single ParB complexes to find 'single chromosomes'. Am I missing something?

(1) The reviewer is correct in that the ParB amount does not correlate to the DNA amount, e.g., a lower amount of ParB can still result in the same level of condensation. This nonintuitive finding fits well with current models of ParB DNA-condensation via transient bridges, as discussed in our recent review, see Tišma et al. *FEMS Microbiology Reviews*. (2023). Here it was discussed that, interestingly, the DNA condensate can remain the same size while the ParB amount fluctuates within it. In other words, more proteins do *not* necessarily result in a larger DNA cluster. For reference to the primary literature, see an extensive discussion of this effect in Tišma et al. *Nucl. Acids Res*. (2023). We now describe this also more clearly in the main text.

(2) Indeed, a higher amount of SMC proteins appears to correlate with more local DNA condensation. This can be naturally attributed to loop extrusion by SMC proteins. We added the additional interpretation of these results in the main text.

On the final point, we stress that these chromosome shapes (crescent, multilobed, dispersed..) were analyzed *before* the selection for single chromosomes (Fig. 1), as mentioned in the text and in point 2 above. *After* we select for the single chromosomes, most cells show crescent shapes (~60%) – which are analyzed in Fig. 2 and now shown in new Fig. S11 (not Supp Fig. S3).

5. Original Question:

The claim that in 67% of cells, >40% of the entire genome was clustered within 500nm of oriC is very interesting. This needs to be confirmed. For instance, strains with fluorescently labeled chromosomal loci away from ori (i.e., using tetO-TetR arrays) can be used to show that indeed they localize near ori.

Answer:

We thank the reviewer for this suggestion to confirm our quantitative claim for origin condensation. The same was pointed out by Reviewer 1 (comment 3). We now provided confirmation of this using our newly constructed strain BSG4610 (Genotype: *1A700, ParB-***mScarlet::**kan**, * amyE***::**Phypherspank-opt.rbs-sirA* *(spec), * hbsU-mTorquais***::CAT, trpC2*), which is now included in Fig. S10. We imaged these cells under the same conditions. The data showed the same crescent chromosomal structure (Fig. S11), and the origin cluster analysis showed similar condensation levels as in our initial SytoxGreen analysis (Fig. 2F-J). In this strain we see 63% and 65% (two replicates) of the cells having more than 40% of their genome withing the origin cluster.

Comment:

How do they know that these cells are not replicating? Selection of a single ParB locus does not ensure lack of replication. If these cells started replicating but did not start replicore separation then their estimate of the % of genomic DNA in the origin cluster would be off.

We have now performed two additional assays that support our conclusions that the strain BSG4610 is *not* replicating in the presence of IPTG (expressed SirA). Plating assays show that the strain's growth is largely halted when IPTG is added, while the BSG4610 strain grows identical to the wild type in the absence of IPTG. This is the same as shown for strains BSG4595 (Fig. S6) and BSG5522 (Fig. S10). Additionally, we performed a qPCR analysis to show the ori:ter ratio in this strain in the absence and presence of IPTG. A strong replication halt, evidenced by an ori:ter ratio of ~1, was observed, as can be seen in Fig. S12B. These data conspire good confidence that most cells have a single chromosome.

Reviewer #3

This re-submission has been improved and most of my comments have been addressed, except for the major one.

Major:

Hi-C is the ideal technique to confirm the authors' exciting findings that Bacillus cells may organize their chromosome in a crescent shape. The results are exciting but they need to be confirmed by a second method, especially when the method is available (Hi-C). The reasoning provided by the authors why they cannot perform this experiment is remarkably weak ("lost the person that was doing Hi-C").

We like to point out that Hi-C would provide us with a contact map, which however does not directly present a three-dimensional structure. That would need extensive coarse-grained modeling from the Hi-C map – akin to the study of Marbouty et al. Mol Cell (2015). This would require an extensive combined experimental/modelling HiC study to infer to the internal 3D chromosomal shape from a contact plot. Following the editorial suggestion, we addressed all other comments and improved the discussion to cover this point.

Minor:

1. Figure S10B. Please include a population analyses, quantification. What percent of cells display the ori and ter at the ends as displayed in S10B? 100%?

The population analysis is now included in the Fig. S10D-E and we added a sentence clarifying the findings in the main text. DNA density is if found almost exclusively between the origin and terminus labels, and *ter* is only in some instances entering the bulk of the chromosome.

2. Given that flow cytometry is not feasible to confirm that cells only have one chromosome, did the authors consider performing simple qPCR for ori-to-ter ratio analyses? This method will work regardless of cells forming chains.

We thank the reviewer for the suggestion and now implemented a qPCR analysis included in Fig. S7 for the strain BSG4595 (and additionally similar results in the Fig. S12 for strain BSG4610). Gratifyingly, the quantified ratios very closely match the ratios previously obtained in the reviewer's request to quantify ori:ter ratio fluorescently using dual-labelled strain (see Fig. S7D).

REVIEWERS' COMMENTS

Reviewer #2 (Remarks to the Author):

My comments have been addressed satisfactorily.

Reviewer #3 (Remarks to the Author):

Given that authors cannot come up with a secondary method to confirm their crescent shape observation of the *B. subtilis* chromosome, this limitation to their findings should be clearly acknowledged in the abstract and throughout manuscript.

The title of the manuscript should include "in the absence of confinement" or something along those lines to avoid misleading readers.

I am confused about this comment from the authors "as our crescent structure matches previously simulated coarse-grained *B. subtilis* chromosomes (Marbouty et al. Mol Cell 2015)...". In the literature, has anyone predicted the crescent shaped chromosome in *B. subtilis* prior to this work? If yes, this should be clearly acknowledged.

Point-by-Point reply to the Reviewers comments

In black – reviewers' comments

In blue – authors' responses

Reviewer #2

My comments have been addressed satisfactorily.

We thank the reviewer for the detailed and useful comments that improved our manuscript over the course of the revision process.

Reviewer #3

Given that authors cannot come up with a secondary method to confirm their crescent shape observation of the *B. subtilis* chromosome, this limitation to their findings should be clearly acknowledged in the abstract and throughout manuscript.

This is now clearly added in the Discussion and Introduction section of the manuscript as a new paragraph, where we outline the limitations of this study and proposed future studies that would further elucidate the chromosome organization.

The title of the manuscript should include “in the absence of confinement” or something along those lines to avoid misleading readers.

The title has now been edited according to the editorial suggestions (and in line with this comment) and the specific notion of unconfined chromosomes has been addressed throughout the manuscript (in Abstract, Introduction, and Discussion).

I am confused about this comment from the authors “as our crescent structure matches previously simulated course-grained *B. subtilis* chromosomes (Marbouty et al. Mol Cell 2015)...”. In the literature, has anyone predicted the crescent shaped chromosome in *B. subtilis* prior to this work? If yes, this should be clearly acknowledged.

This is clearly acknowledged throughout the manuscript and the comparison to Marbouty et al work (which is cited 15 times in the main text) is described in detail in the discussion section.